# Learning *cis*-regulatory principles of ADAR-based RNA editing from CRISPR-mediated mutagenesis

Xin Liu [1,7], Tao Sun[1,7], Anna Shcherbina[2,7], Qin Li [1], Inga Jarmoskaite [1], Kalli Kappel [3], Gokul Ramaswami[1], Rhiju Das [4,5], Anshul Kundaje [1,6✉] & Jin Billy Li [1✉]

Adenosine-to-inosine (A-to-I) RNA editing catalyzed by ADAR enzymes occurs in double-stranded RNAs. Despite a compelling need towards predictive understanding of natural and engineered editing events, how the RNA sequence and structure determine the editing efficiency and specificity (i.e., *cis*-regulation) is poorly understood. We apply a CRISPR/Cas9-mediated saturation mutagenesis approach to generate libraries of mutations near three natural editing substrates at their endogenous genomic loci. We use machine learning to integrate diverse RNA sequence and structure features to model editing levels measured by deep sequencing. We confirm known features and identify new features important for RNA editing. Training and testing XGBoost algorithm within the same substrate yield models that explain 68 to 86 percent of substrate-specific variation in editing levels. However, the models do not generalize across substrates, suggesting complex and context-dependent regulation patterns. Our integrative approach can be applied to larger scale experiments towards deciphering the RNA editing code.

[1] Department of Genetics, Stanford University, Stanford, CA, USA. [2] Department of Biomedical Data Science, Stanford University, Stanford, CA, USA. [3] Biophysics Program, Stanford University, Stanford, CA, USA. [4] Department of Biochemistry, Stanford University, Stanford, CA, USA. [5] Department of Physics, Stanford University, Stanford, CA, USA. [6] Department of Computer Science, Stanford University, Stanford, CA, USA. [7] These authors contributed equally: Xin Liu, Tao Sun, Anna Shcherbina. ✉email: akundaje@stanford.edu; jin.billy.li@stanford.edu

RNA editing greatly diversifies the transcriptome and proteome in higher eukaryotes[1,2]. In animals, the predominant type of RNA editing is the hydrolytic deamination of adenosine (A) to form inosine (I), catalyzed by adenosine deaminase acting on RNA (ADAR)[3,4]. Abnormal A-to-I RNA editing is strongly linked to autoimmune diseases, neurological disorders and cancers[5,6]. Humans have two catalytically active ADAR proteins, ADAR1 and ADAR2, responsible for the editing of millions of RNA editing sites[7,8]. Adenosines in perfect or nearly perfect dsRNA duplexes, formed mainly by inverted repeats, are promiscuously edited[9]; in contrast, adenosines in imperfect dsRNA structures, can be edited by ADARs with high specificity and efficiency[10]. How RNA editing is regulated to determine its efficiency and specificity is poorly understood. Both the primary sequence and secondary structure (i.e., cis-acting regulatory elements) have been proposed to regulate ADAR editing[4,11–16]. A preferred sequence motif has been defined, including the 5′ and 3′ nearest-neighboring positions (−1 and +1 nt) to the editing site[11–13]. Editing can be enhanced or suppressed by deviations from perfect base-pairing (i.e., mismatches, bulges, and loops), suggesting complex structural contributions to editing specificity[11–13,17]. The quantitative trait loci (QTL) mapping approach has been used to identify genetic variants associated with variability in RNA editing in Drosophila and humans, suggesting that many editing QTLs act through changes in the local and distal secondary structure for edited dsRNAs, consistent with the importance of RNA structure[18,19]. However, previous studies have generally been limited to small numbers of natural or engineered variants. We lack the systematic sequence and structure variations required for the development of predictive models of editing, therefore a high-throughput, systematic mutagenesis approach is needed.

Here, we combine CRISPR/Cas9 genome engineering, next-generation sequencing, and machine learning to decipher cis-regulatory RNA sequence and structural elements that affect ADAR-mediated RNA editing. As proof-of-concept, we choose three representative RNA editing substrates and introduce hundreds of mutations at the endogenous loci in human cells, using the CRISPR-mediated approach. We use supervised machine learning to build predictive models of substrate-specific RNA editing levels based on a variety of cis-sequence and structural features. We identify highly edited structures different from wild-type (WT) structure (referred to as alternative structures), and general, as well as idiosyncratic features that determine editing efficiency of individual substrates, highlighting the complexity of the cis-regulatory editing code. Our integrative approach, named predicting RNA editing using sequence and structure (PREUSS), lays the foundation for developing predictive models of RNA editing.

## Results

### CRISPR/Cas9-mediated mutagenesis to interrogate endogenous RNA editing

To interrogate the effects of cis-regulatory elements of RNA editing, we applied the CRISPR/Cas9 technology (Fig. 1a) to introduce mutations at the endogenous loci of three natural ADAR1 substrates (NEIL1, TTYH2, and AJUBA; Fig. 1b, see "Methods"). The mutations were introduced both in the strand containing the editing site ("editing strand") and in the complementary sequence involved in forming the secondary structure, which we refer to as editing complementary sequence (ECS). Briefly, we designed CRISPR guide RNAs (gRNAs) targeting the regions of interest, as well as oligonucleotide donors carrying mutations to direct knock-in (KI) mutations through the CRISPR/Cas9-mediated homology-directed repair (HDR) pathway[20]. To measure the RNA editing levels of the resulting variants, we performed targeted amplicon deep sequencing.

Because the variant and the associated editing site are in the same transcript, there is no need to perform laborious clonal selection for homozygotes of the variants. Because each designed variant has a unique sequence, we successfully performed large-scale multiplex mutagenesis and measured the editing levels without the aid of barcodes (see "Methods" for details).

In a pilot experiment to introduce one or more mutations near the editing site, we used a single degenerate donor oligonucleotide with mutations at each position to randomize the region from −3 to +3 positions of the editing strand for NEIL1 (Fig. 1c) and a 10 nt region on the ECS of TTYH2 (Supplementary Fig. 1a). These random mutations provide a rapid means to evaluate the CRISPR/Cas9 KI efficiency and the effects of mutations[10]. We observed that three or more mutations almost always lead to an abolishment of editing events (Fig. 1d, e and Supplementary Fig. 1b). Therefore, to generate variants that lead to a wide range of editing levels, we next performed targeted mutagenesis using a pool of 200–300 donor oligonucleotides with designed mutations, focusing on single and double mutations with larger mutagenesis regions around the editing site in the editing strand and the ECS (Figs. 1b and 2a, and Supplementary Fig. 3a–c). For NEIL1 and TTYH2, we designed all possible single mutations both in the editing strand and the ECS (with the exception of −1 and +1 positions for NEIL1 where A-to-G mutations, would be indistinguishable from A-to-I editing). For AJUBA, we only designed mutations in the editing strand due to its long-range ECS. We selectively designed a subset of double-transversion mutations (11% all possible double mutations) that theoretically disrupt the original base-pairing at the mutation position. For NEIL1, we also introduced compensatory mutation variants that theoretically maintain base-pairing. In addition, we designed indel variants to study the effects of selected secondary structure features of NEIL1, such as bulges, internal loops, and stem length (RNA sequences and editing levels see Supplementary Data 7).

Overall, we achieved high KI efficiency as 10–20% of the sequenced RNAs at the target locus carried mutations, similar to previous reports for a similar approach[20]. We were able to reliably detect >90% of our designed variants after using stringent quality control filters. The KI results and editing measurements were highly reproducible (Fig. 1f and Supplementary Fig. 1c–f). Interestingly, we discovered that similar KI efficiency and editing results were achieved when using ssDNA oligonucleotides or dsDNA (e.g., PCR product) as the donor for CRISPR-mediated HDR (Supplementary Fig. 1g–j). Using dsDNA PCR products greatly simplified the procedures and reduced the cost of the experiments. The coverage of RT-PCR product for each variant was generally well correlated with the corresponding coverage of the product amplified from gDNA ($R^2 = 0.87$ for NEIL1 and $R^2 = 0.25$ for TTYH2; Supplementary Fig. 2a, d), suggesting that the RNA abundance is generally not affected by the introduced variants. There is no correlation between the RNA or gDNA coverage with the editing level for all three substrates, which is consistent with previous reports[21,22] that argues against potential influence of the substrate expression level on the editing level (Supplementary Fig. 2b, c, e–g).

### Intertwined effects of primary sequence and secondary structure on editing levels

We compared the effects of single and double mutations in terms of the type (transition or transversion) and location of the mutation across all three RNA substrates (Fig. 2a and Supplementary Fig. 3a–c). We used the computationally predicted minimum free energy (MFE) secondary structure of each RNA variant to dissect the associations between mutations and structure. The mutational effects are summarized for each RNA substrate below.

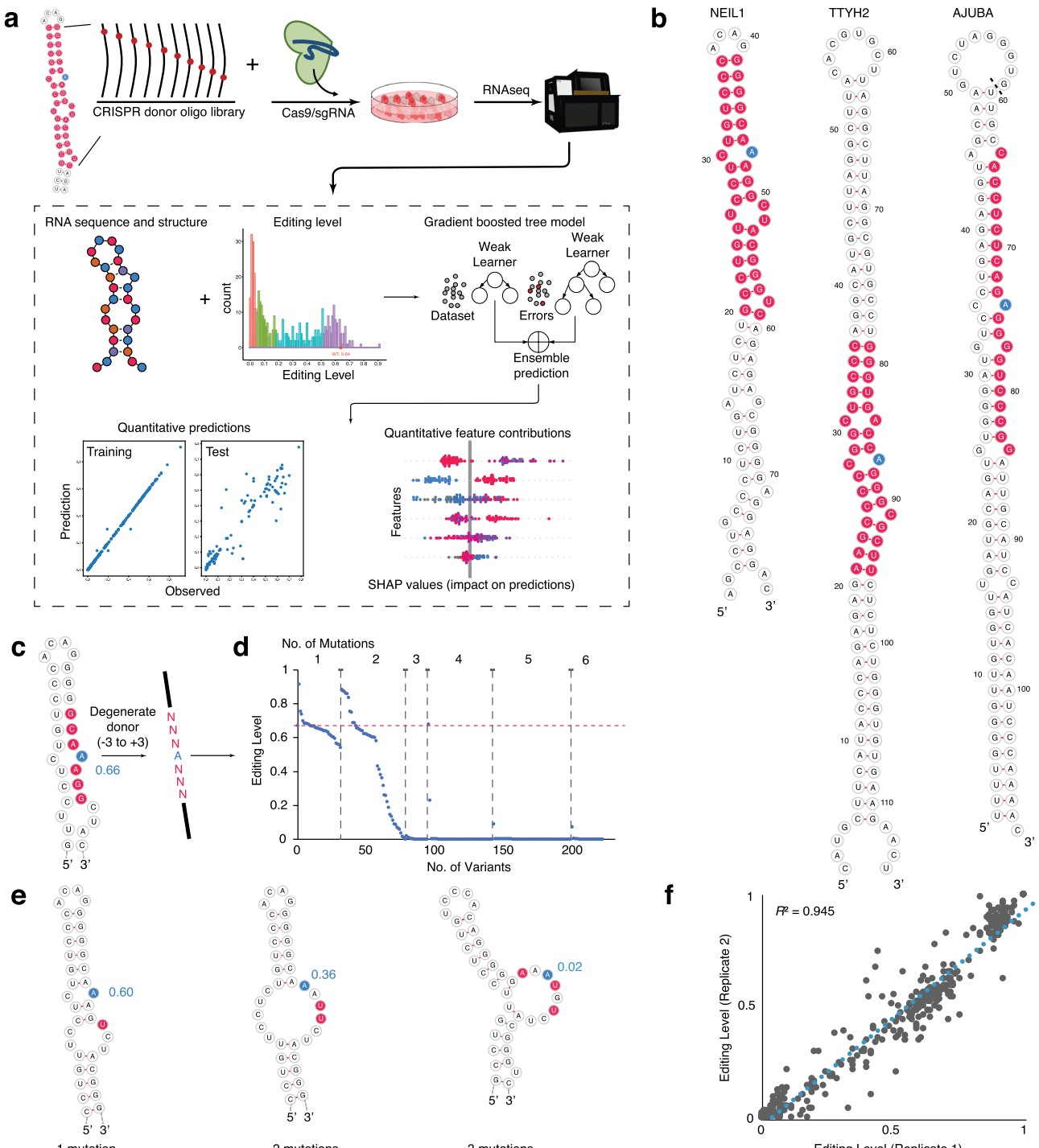

**Fig. 1 CRISPR/Cas9-mediated mutagenesis in endogenous RNA to dissect RNA editing by ADAR1 in cells. a** Overview of the experimental methods and computational pipeline. CRISPR/Cas9-mediated homology-directed repair is applied to mutagenesis of endogenous RNA in HEK293T cells. A supervised machine learning method (a gradient boosted tree, XGBoost) was applied to develop quantitative models that predict how *cis*-elements, such as RNA sequence and secondary structure determine RNA editing level. **b** Sequence and secondary structure of the three RNAs, NEIL1, TTYH2, and AJUBA, for targeted mutagenesis. The residues subjected to mutations are highlighted in red and the specific editing site is in blue. For AJUBA, partial sequences from the genomic sequences are taken to focus on the region of interest. Therefore, the G59 and U60 shown in **b** is 524 nt apart in the genomic region. **c** Degenerate donor oligos are designed for the −3 to +3 nt region around the specific editing site in the NEIL1 substrate. The mutagenized region is highlighted in red and the editing site in blue. The value of editing level is shown in blue. **d** The distribution of editing level by the number of mutations from the results of the degenerate NEIL1 library from **c**. **e** Examples of how the number of mutations affect the RNA secondary structure of NEIL1. The mutagenized nucleotied is highlighted in red and the editing site in blue. The value of editing level is shown in blue. **f** Reproducible editing measurement of the two replicates of the targeted mutagenesis library of NEIL1 shown by pairwise comparison with Spearman $R^2$ labeled.

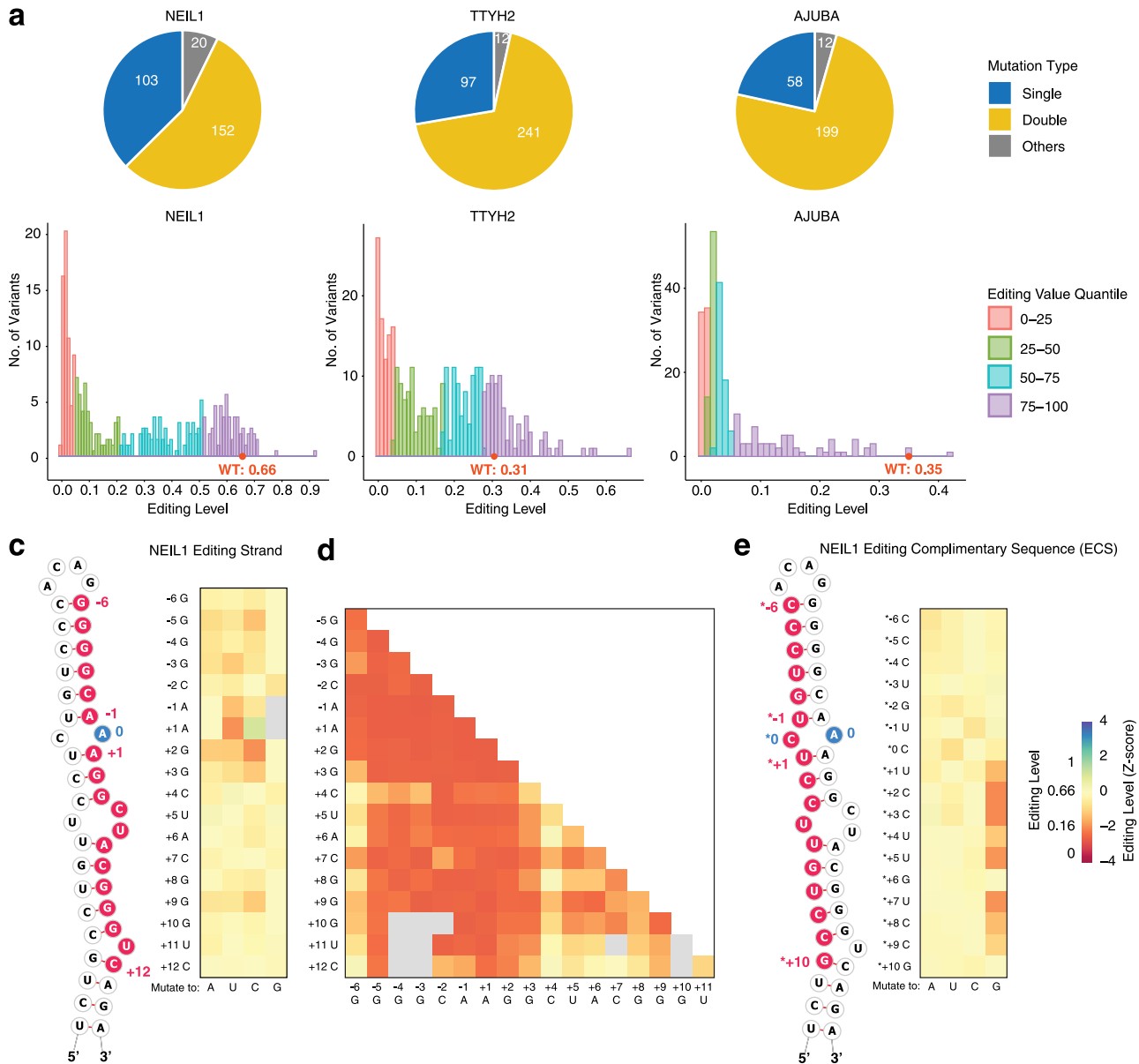

**Fig. 2 RNA editing results from the targeted mutagenesis experiments. a** Number of the types of mutations made in each targeted mutagenesis library, including single mutations (blue), double mutations (yellow), and other mutants such as indels (gray). **b** Distributions of editing levels for each targeted mutagenesis library, colored by editing level quantile in each RNA library. Pink, 25% quantile; green, 25–50% quantile; blue, 50–75% quantile; purple, 75–100% quantile. **c**, **d** Heatmap of editing levels from **c** single and **d** double mutations in the editing strand of NEIL1. **e** Heatmap of editing levels from single mutation in the editing complementary sequence (ECS) of NEIL1. Editing level of WT NEIL1 is 0.66 ± 0.06. The Z-score is calculated as described in "Methods" and the WT editing level Z-score is 0. **c**–**e** shares the same heatmap color scale shown in **e**, reflecting average editing level from six biological replicates. In **c** and **e**, the mutagenized region is highlighted in red and the editing site in blue in the partial illustration of the secondary structure of NEIL1 RNA.

*NEIL1.* For NEIL1, most single mutations led to minor decreases in editing ($-1 < Z$-score $< 0$), with the largest effects observed at positions +1 and +2 relative to the editing site (Figs. 2c and 3b). Exceptions from this pattern were the large effects ($Z$-score $< -1$) of G mutations downstream from the editing site in the ECS strand. An example of how a G mutant changes secondary structure is shown in Supplementary Fig. 4d. Some RNA variants have the same predicted RNA structure but different editing level, most simply suggesting a primary sequence effect (Supplementary Fig. 4e). To decouple sequence and structural effects at each position, we considered six categories of single mutations. The simple "transition" (i.e., purine to purine, pyrimidine to

pyrimidine) and "transversion" (pyrimidine to purine and vice versa) categories indicate that the RNA structure was unchanged compared to the WT. "Transition + break" and "transversion + break" categories indicate that the mutation disrupted the base-pair at the mutation site, and the "transition + shift" and "transversion + shift" categories include all other scenarios, such as the formation of a new base pair or disruption of more than one base pair (Fig. 3a). We observed moderate effects ($Z$-score $< -1$) for the transversion mutations that also disrupted base-pairing (transversion + break) or caused other structural changes (transversion + shift) at positions in close vicinity to the editing site ($-5$, $-1$, +1 to +3, +9), and at the 3′ side of the editing site

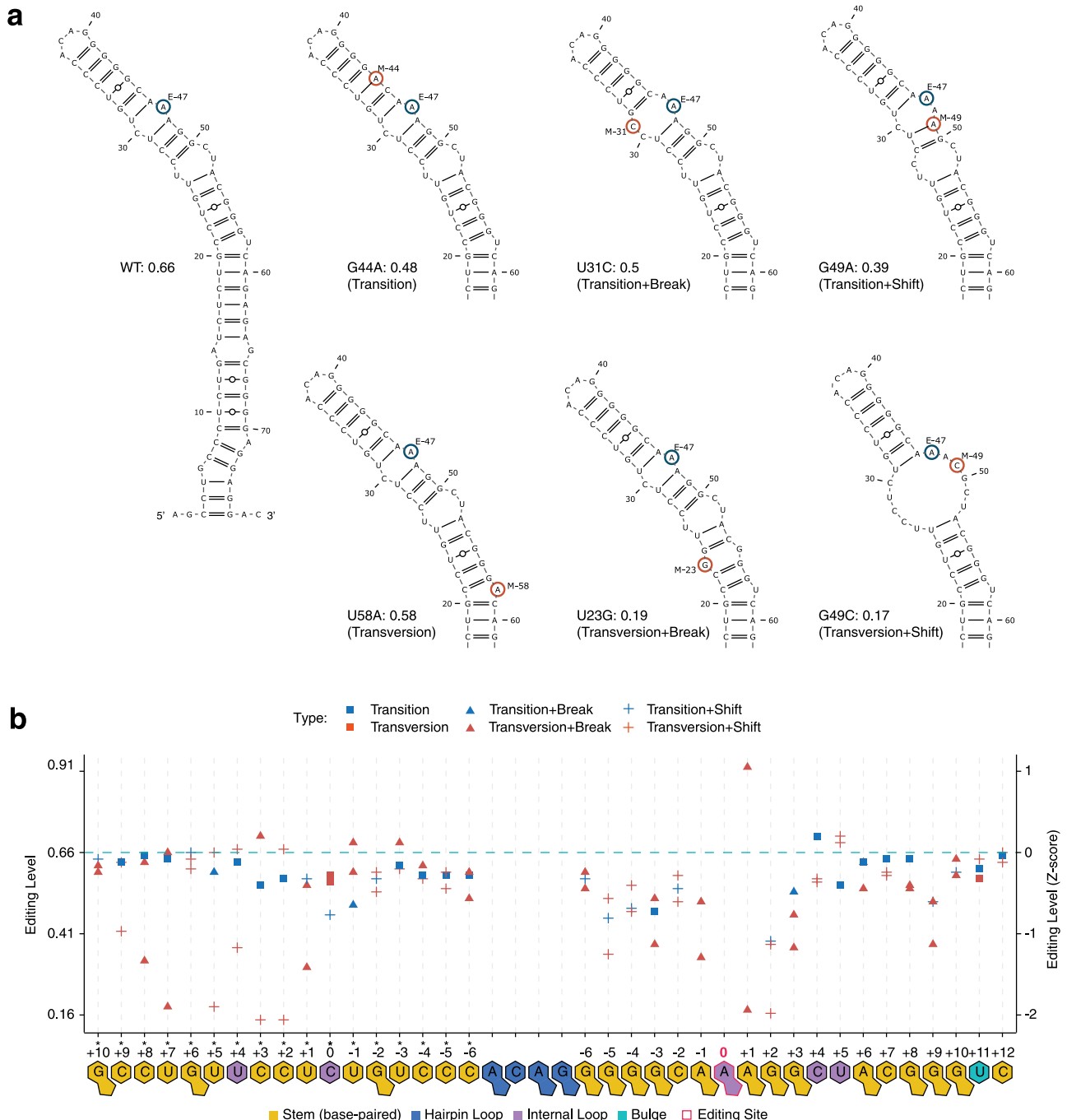

**Fig. 3 Effects of NEIL1 single mutations on RNA structure. a** Editing site is indicated by blue circle and the mutation site is marked by red circle. The editing level shown is the average value from six biological replicates. The single mutations were grouped into six types: sequence change (transition or transversion) without change in the structure, the sequence change (transition or transversion) resulted in breaking of the base pair at the mutation site (break), or resulted in breaking more than one base pair or forming of new base pair(s) (shift). E editing site, M mutation site. **b** Position-specific effects of NEIL1 single mutations, categorized by six types: transition mutation that does not affect RNA secondary structure (transition, blue square), transition mutation that disrupt the base pair at the mutation site (transition + break, blue triangle), transition mutation that leads to disruption of more than one base pair and/or formation of new base pair (transition + shift, blue cross), transversion mutation that does not affect RNA secondary structure (transversion, red square), transversion mutation that disrupt the base pair at the mutation site (transversion + break, red triangle), transversion mutation that leads to disruption of more than one base pair and/or formation of new base pair (transversion + shift, red cross). The $Z$-score is calculated for the NEIL1 RNA library as described in "Methods" and the WT editing level $Z$-score is 0.

(*+1 to *+8). Double-transversion mutations in the editing strand of NEIL1 had overall pronounced effects on editing (Fig. 2d), and the strongest effects ($Z$-score < −2) were observed when at least one of the mutations was in close proximity to the editing site (positions −6 and +2). The effect was generally

smaller ($Z$-score > −2) if one of the mutations was in a non-base-paired region (+4, +5, +11; Fig. 2d).

*TTYH2.* In contrast to NEIL1, where the vast majority of mutations decreased editing, a large proportion of TTYH2 single

mutations increased the editing efficiency ($Z$-score > 0; Supplementary Figs. 3a, b and 4a). This difference may be explained by the lower WT editing level for TTYH2 (0.31) than for NEIL1 (0.66). Similar to NEIL1, single mutations closer to the editing site (−2 to +1, *−3 to *+4) tended to have larger negative effects on editing levels ($Z$-score < −1; Supplementary Fig. 4a). Interestingly, several single mutations located both upstream (−6 to −3) and downstream (+6 to +8, *+6) of the editing site increased editing levels ($Z$-score > 0). For TTYH2 double mutations, the effects were most negative when at least one mutation was located around the editing site (−2 to +6 in the editing strand and *−3 to *+5 in the ECS; Supplementary Fig. 3a, b).

*AJUBA.* We only examined the mutations on the editing strand of AJUBA. In contrast to NEIL1 and TTYH2, many single mutations of AJUBA were sufficient to disrupt editing. Also, all double mutations abolished editing regardless of positions (Supplementary Fig. 3c). Most of the "transition + break", "transition + shift", and the "transversion + shift" single mutations had large effects ($Z$-score < −2; Supplementary Fig. 3b). Many AJUBA single mutations have much larger effects ($Z$-score < −3) than single mutations in NEIL1 and TTYH2. This difference may be explained by the long distance (524 nt) between AJUBA editing strand and ECS, such that mutations may lower the probability of forming this long-range structure relative to alternative proximal structures; alternatively, primary sequence might have a larger influence on editing of the AJUBA RNA.

Taken together, these results are consistent with previous observations of intertwined sequence and structure effects on editing. These effects also vary among the three different RNA substrates, suggesting substrate-specific *cis*-regulation rules.

**RNA structural features affect editing levels**. Next, we systematically explored the effects of changes to the RNA secondary structure on editing levels. We found that compensatory double mutations in NEIL1 that did not affect secondary structure resulted in only minor reduction of editing levels (Fig. 4a–c). To investigate how a specific structural change affects editing efficiency, we designed several indels that change the predicted secondary structure of NEIL1 (Fig. 4d). Shortening the 5′ stem or breaking base-pairs within this stem abolished editing, suggesting the importance of this region for editing; increasing the stem length by 2-bp did not increase editing efficiency (Fig. 4d and Supplementary Fig. 4f). The 3′ base-pairing is also critical because breaking it led to nearly complete disruption of editing (Supplementary Fig. 4f). When we replaced the downstream 3′ internal loop with either a canonical base-pair or wobble base-pair, the editing efficiency decreased by 50% ($Z$-score = −1), suggesting the importance of this structural feature (Fig. 4d and Supplementary Fig. 4f). Enlarging the loop with additional nucleotides resulted in mild (−1 < $Z$-score < 0) reductions in editing levels (Fig. 4d). As expected, editing site structures containing an A:C mismatch (1:1 internal loop) exhibited higher editing levels on average than when the editing site resided in a larger loop ($P$ < 0.0001 by Wilcoxon test, Supplementary Fig. 4g). However, several editing site structures harboring non-A:C mismatches also showed strong editing levels for NEIL1 and TTYH2 (Supplementary Fig. 3d, e), indicating that additional factors affect editing efficiency.

We reasoned that structural and thermodynamic features affecting RNA stability could also affect editing efficiency[15]. We observed significantly greater predicted structural stability in highly (highest 25 percentile of editing level in each RNA library) compared to lowly (lowest 25 percentile) edited NEIL1 ($P$ < 0.0001) and AJUBA ($P$ < 0.001) variants, based on both the MFE

structure (Fig. 5a) and the predicted structural ensemble (Fig. 5b). We also observed significantly higher MFE frequency for NEIL1 ($P$ < 0.01) and TTYH2 ($P$ < 0.0001; Fig. 5c) and lower ensemble diversity for NEIL1 ($P$ < 0.0001; Fig. 5d) in highest edited quantile. The same observation held when stability was approximated by the number of base-pairs formed (Fig. 5e).

We hypothesized that RNA variants that are structurally more similar to the WT would result in editing levels similar to WT. We quantified structural similarity using two measures: the probability of active conformation, which indicates the probability of forming WT-like secondary structure in the predicted structural ensemble for each RNA variant (Fig. 5f) and a similarity score that indicates the degree to which the MFE structure of a variant is similar to the MFE structure of WT (scale is 0–1, from least similar to identical structure, Fig. 5g). A higher probability of active conformation was observed in the highly edited variants compared to lowly edited variants for all three substrates ($P$ < 0.0001, Fig. 5f). We found significant differences ($P$ < 0.0001) of similarity score between highly and lowly editing variants for NEIL1 and AJUBA (Fig. 5g).

However, when we considered all variants across the entire editing spectrum, instead of highly versus lowly edited variants, no significant correlations were observed between individual features and RNA editing levels (Supplementary Figs. 5 and 6). These results show that individual sequence, structure, and stability features of variants can explain the differences between highest and lowest edited substrates, but only have limited predictive association with quantitative editing levels. Therefore, we decided to carry out integrative analyses of RNA sequence and structure features to model quantitative editing levels.

**RNA clustering reveals alternative structures that support efficient editing**. Given that no single property of the RNA substrates correlated strongly and consistently with editing efficiency (Supplementary Figs. 5 and 6), we used machine learning to dissect the collective effects of different features on editing. First, we performed a hierarchical clustering analysis based on variant sequence and structure. We clustered the NEIL1 and TTYH2 libraries because the editing levels are widely distributed compared to the AJBUA results. We used the locARNA pipeline[23,24] that takes into account both the sequence and the MFE structure (Fig. 6a). Because the sequence variation is relatively small, the similarity and difference among the MFE structures was weighted highly for the resulting hierarchical clustering. The resulting clusters of RNA variants generally share a similar core structure and show similar editing levels (Fig. 6b and Supplementary Figs. 7–9). Interestingly, we found clusters of RNA with predicted structures distinct from WT (alternative structures) that are edited with near-WT efficiencies, both for NEIL1 (e.g., clusters 2, 4, and 8 in Fig. 6), and for TTYH2 (e.g., clusters 3, 2, 7, and 8 in Supplementary Figs. 8 and 9). As an example, positioning of the NEIL1 editing site in an asymmetric 2:3 internal loop (cluster 8 in Fig. 6b) instead of the 1:1 A:C loop seen in WT (cluster 1 in Fig. 6b) appears to maintain, and even enhance, the editing efficiency. In contrast, a 1:2 internal loop in cluster 7 (Fig. 6b) with similar downstream structures as cluster 8 is mostly poorly edited. Cross-cluster comparisons further illuminate the contributions of certain structural features to editing. For example, comparing NEIL1 cluster 5 with cluster 1 (Fig. 6) suggests a negative effect of a bulge in the 5′ stem. While NEIL1 prefers a good 5′ stem structure, the TTYH2 can tolerate symmetric internal loops in the 5′ stem (Supplementary Figs. 8 and 9).

**Machine learning models accurately predict substrate-specific RNA editing levels from sequence and structure features**. To quantitatively capture the complex relationship between editing

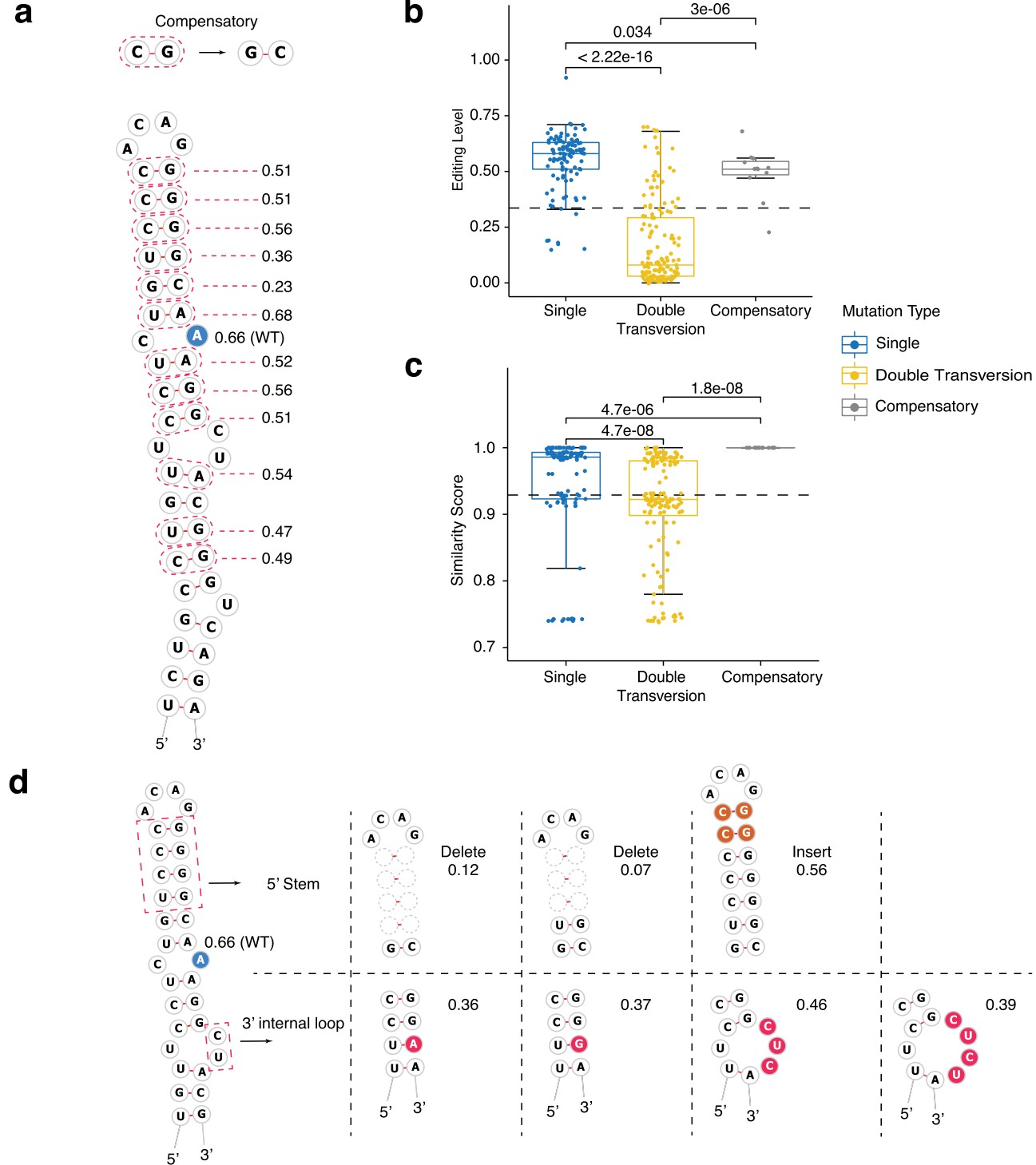

**Fig. 4 Examples of RNA secondary structure changes of NEIL1 variants. a** Compensatory mutation generally maintains a high editing level. Editing site is highlighted in blue. The dashed circle marks the location of compensatory double mutation. The **b** editing level and **c** similarity score (normalized score calculating the similarity of the MFE structure of each variant to the WT) vary by different mutation types. Single mutation (blue dot); double-transversion mutation (yellow dot); compensatory mutation (gray dot). The data points shown are the average editing level from six biological replicates. Boxplot: center line, median; box limits, upper and lower quantiles; whiskers, ±1.5× interquartile range (IQR). The *P* values from two-sided Wilcoxon rank-sum test are shown on each test set. **d** Alterations in the 5′ stem and 3′ non-stem structure elements affect editing level. Editing site is highlighted in blue and mutation region shown as dashed circle (deletion of stem from the 5′ stem), orange highlight (insertion of stem), or red highlight (mutation, deletion, or insertion of nucleotides of the 3′ internal loop).

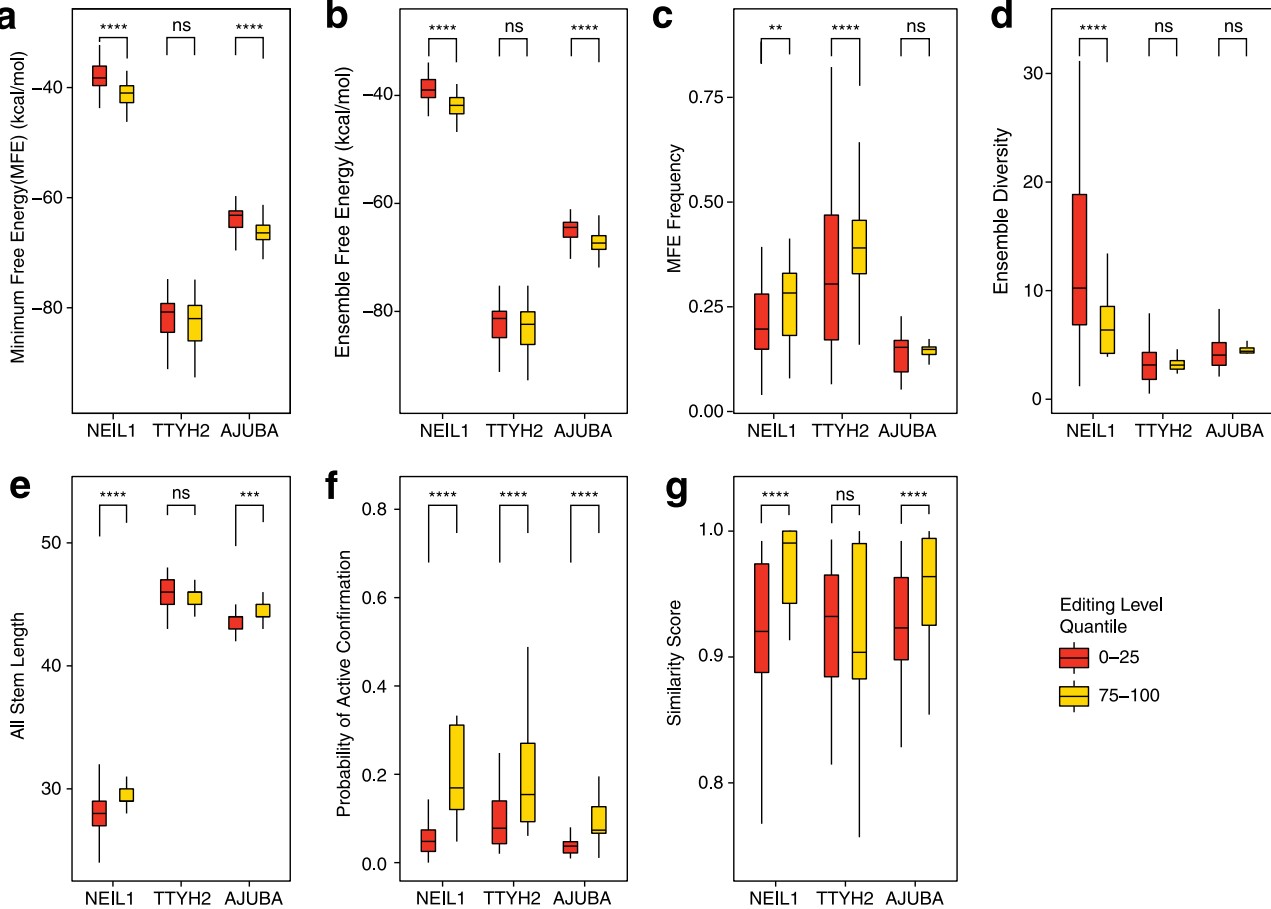

**Fig. 5 Cis-regulatory features explain differences of editing levels among RNA variants. a–g** Comparing the difference of the highly edited (75–100 percentile in editing level in the library, yellow box) with the lowly edited (0–25 percentile, red box) variants in each RNA library in terms of thermodynamic and structural features. Two-sided Wilcoxon rank-sum test: ns nonsignificant; **$P < 0.01$; ***$P < 0.001$; ****$P < 0.0001$, and the exact $P$ values for each feature are: **a** minimum free energy (MFE), $P$ values: NEIL1 = 4.3e−12, TTYH2 = 0.44, AJUBA = 9.3e−07; **b** ensemble free energy, $P$ values: NEIL1 = 2.2e−12, TTYH2 = 0.67, AJUBA = 7.2e−07; **c** MFE frequency, $P$ values: NEIL1 = 0.0013, TTYH2 = 6.8e−05, AJUBA = 0.8751; **c** MFE frequency, $P$ values: NEIL1 =, TTYH2 =, AJUBA =; **d** ensemble diversity, $P$ values: NEIL1 = 9.3e−05, TTYH2 = 0.72, AJUBA = 0.19; **e** all stem length, $P$ values: NEIL1 = 4.6e−09, TTYH2 = 0.81011, AJUBA = 0.00024; **f** probability of active conformation, $P$ values: NEIL1 = < 2e−16, TTYH2 = 1.3e−08, AJUBA = 3.8e−11; **g** similarity score, $P$ values: NEIL1 = 1.3e−09, TTYH2 = 0.87, AJUBA = 2.5e−06. Boxplot: center line, median; box limits, upper and lower quantiles; whiskers, ±1.5× IQR. The editing level are the average editing level from six biological replicates.

levels and multidimensional RNA sequence and structure features, we turned to machine learning models. A set of 125 features were derived to annotate the RNA variants (see "Methods" and Supplementary Datas 1–4 for feature annotations for all variants of NEIL1, TTYH2, and AJUBA, respectively). The sequence features summarize various properties of the primary RNA sequence of each variant at and around the vicinity of the editing site where the mutations were made. We used the bpRNA[25] tool to assign all residues in each variant to diverse structural elements, such as hairpin loops, bulges, internal loops, stems, multi-loops and closing pairs (Fig. 7a). We chose to featurize the bpRNA structural annotations at the editing site and adjacent regions (up to 3 bpRNA structural elements upstream and downstream from the editing site structure element, detailed in Supplementary Data 1; Fig. 7b), as these regions within the RNA substrate fully encompass the interaction site with the ADAR deaminase domain (Fig. 7c)[26,27]. The 125 features were further grouped into nine major categories for purposes of feature interpretation (Supplementary Data 1). Gradient boosted trees (GBTs) were trained via the XGBoost algorithm[28]. We trained and tuned GBTs on distinct subsets of RNA variants to map their feature annotations to corresponding real-valued editing levels or

binarized labels obtained by thresholding editing levels into two classes (edited versus not edited).

First, we evaluated the prediction performance of our model for each substrate. We trained and tuned models on a subset of variants and then tested model performance on a held-out test set of variants of the same substrate. For NEIL1, the models accounted for 85.6% of the variance ($R^2$) in ADAR editing levels for variants in the held-out test set, with a Spearman correlation ($R_s$) of 0.92 between observed and predicted editing levels. Binary editing status was also predicted accurately (area under precision-recall curve, auPR = 0.97). Similarly, high test set predictive performance was obtained for TTYH2 variants ($R^2 = 0.68$, $R_s = 0.91$, auPR = 0.81) and AJUBA variants ($R^2 = 0.79$, $R_s = 0.90$, auPR = 0.93). Augmenting the training set for each substrate with variants from the other substrates did not result in any significant improvements in model performance (Supplementary Data 5 and Supplementary Fig. 10). These results indicate that it is possible to predict RNA editing levels of new mutations in a substrate with high accuracy from sequence and structure features, using integrative machine learning models trained on a subset of mutations from the same substrate (Fig. 7d).

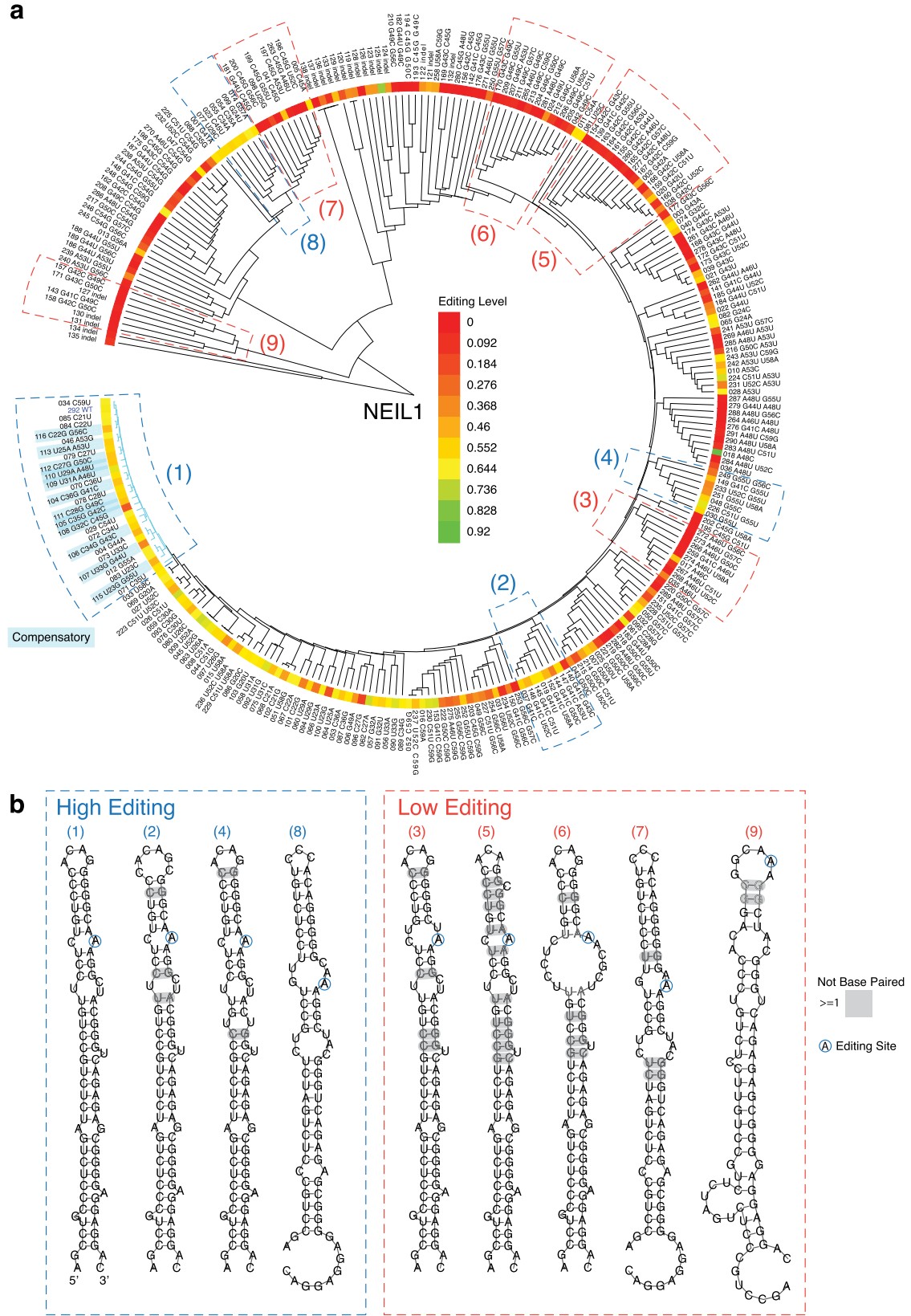

**Fig. 6 NEIL1 RNA clustering reveals efficiently edited alternative structures. a** NEIL1 variants are clustered by RNAclust from the multiple sequence–structure alignment generated by mlocarna. The editing level of each variant are shown according to the heatmap scale. The sequence and structure corresponding to each RNA ID are listed in Supplementary Data 2. **b** Consensus secondary structure of selected clusters from **a** and grouped by editing levels. The gray box ("not base-paired") indicates that there is at least one variant within the cluster that has a different MFE structure at this position (see examples in Supplementary Fig. 7).

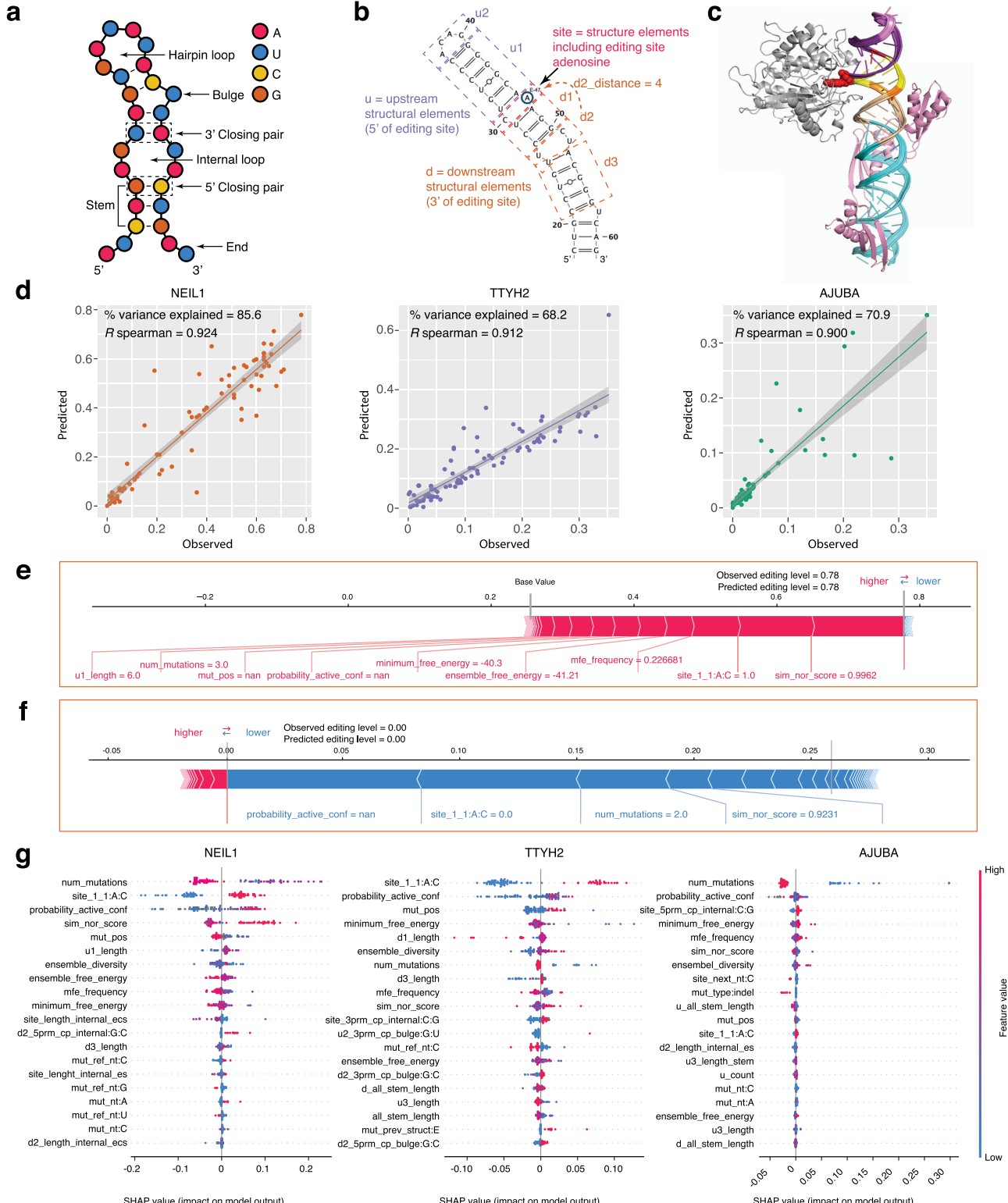

Next, we tested whether models trained on variants of one or more substrates could predict editing effects of mutations in a different substrate. We observed a significant drop in model performance for cross-substrate prediction of RNA editing (Supplementary Fig. 11,). For example, a model trained on NEIL1 variants yielded lower performance on AJUBA variants ($R^2 < 0.05$, $R_s = 0.68$, auPR = 0.69) and TTYH2 variants ($R^2 < 0.05$, $R_s = 0.46$, auPR = 0.59), as compared to a model trained and tested on NEIL1 variants ($R^2 = 0.88$, $R_s = 0.93$, auPR = 0.97).

Similarly, a model trained on all NEIL1 and TTYH2 variants also yielded lower predictive performance when tested on AJUBA variants ($R^2 < 0.05$, $R_s = 0.66$, auPR = 0.29), albeit higher than the model trained on either substrate independently. The same held true for all models trained on two of the substrates and evaluated on the third—lower performance compared to within-substrate training and evaluation.

The inability of our current models to accurately generalize predictions to new substrates is not entirely surprising

**Fig. 7 Quantitative model predicts editing level by combining complex RNA sequence and structure features. a** Structure features annotated by bpRNA and included in featurization of RNA variants. **b** High-level feature groups for input to XGBoost analysis. u1 = structural element immediately upstream (5′) of editing site; u2 = structural element upstream of u1; site = structural element within which the editing site is found; d1 = structural element downstream (3′) of editing site; d2 = structural element downstream of d1; d3 = structural element downstream of d2. Definition of each feature is listed in Supplementary Data 1. **c** Illustration of a putative model for binding of the NEIL1 RNA to the ADAR1. The ADAR1 deaminase domain (silver) are modeled from ADAR2 by Phyre2. The dsRNA-binding domains (pink) are modeled in one possible conformation as described in the "Methods". The editing site mismatch (also considered a 1:1 internal loop) on NEIL1 is shown in red and the editing A shown as space filled. The upstream (purple and light purple) and downstream (yellow, orange, and light orange) immediately adjacent to the editing site are colored according to shown in **b**. **d** XGBoost editing level predictions for variants of NEIL1 (orange), TTYH2 (purple), and AJUBA (green) within the test split (15% random split of positions). $R^2$ is a measure of the % variance explained. Spearman $R$ indicates correlation between observed and predicted editing values. Error bands (in gray) the 95 pointwise confidence bound for the mean predicted value, using linear smoothing. **e** SHAP annotation of feature contributions for the NEIL1 test split variant with the highest observed editing level. Features with positive SHAP scores (drive the prediction over the dataset base value) are indicated in pink; features with negative SHAP values (drive the prediction below the dataset base value) are indicated in blue. Base value refers to the mean predicted editing level across the test split. Output value refers to the XGBoost prediction on this example. The four features with the highest absolute value SHAP scores are shown. **f** SHAP annotation of feature contributions for the NEIL1 test split variant with the lowest observed editing level. **g** SHAP values for the 20 most important features driving XGoost editing level predictions on the test split for NEIL1, TTYH2, and AJUBA. Each dot indicates a variant in the test split and the dot color shows the SHAP value from high (red) to low (blue). Features (y-axis) are ranked from top (most significant) to bottom (least significant) by predictive importance.

considering the diversity of the substrates and the small number (three) of distinct substrates available for model training. It is likely that the challenge of cross-substrate training may be solved by training on a larger number of variants from diverse substrates, and future efforts will focus on this task. However, given the success of our substrate-specific models in predicting editing effects for unseen mutations within each substrate, we decided to interpret these models to investigate the features that may be predictive of RNA editing levels.

**Model interpretation provides insights into common and substrate-specific features associated with RNA editing efficiency.** For each of the three substrate-specific models, we used the TreeExplainer SHAP (SHapley Additive exPlanations) algorithm to quantify the contributions (or importance) of all features to the RNA editing predictions of each variant in the test sets[29]. The SHAP importance score of a feature with a specific value for a variant of a substrate estimates how much the feature contributes to pushing the model's output from a baseline editing level to the predicted RNA editing level for the variant. The baseline editing level is defined as the average editing level across all variants in the test set. Examples of how SHAP scores illuminate feature importance are illustrated in Fig. 7e, f. Figure 7e illustrates the SHAP scores for the five most important features for the NEIL1 test variant (NEIL1 RNA ID 092, U31G) with a high observed editing level (0.78) agrees with model prediction (0.78). For the NEIL1 test set, the baseline (mean) predicted editing value is 0.25. We display the contribution of all feature values for this variant in pushing the prediction from the baseline of 0.25 to the predicted output value of 0.78. The feature "sim_nor_score (same as in Fig. 5g, normalized similarity score comparing MFE structure of variant to WT) = 0.99", is estimated to have the highest importance and increases the prediction of editing level by 0.09 (SHAP value) from the baseline. The contribution of the editing site with an A:C mismatch has a SHAP value of 0.05, and so on. Although a larger number of mutations generally decreases editing level (Fig. 7g), the "num_mutations = 3" has a positive SHAP value (red) for this variant, highlighting the ability of the model to pick up different feature combinations. Conversely, Fig. 7f illustrates how feature values unfavorable to ADAR editing result in a predicted editing level of 0 for another variant (NEIL1 RNA ID 142, G41C/C45G) of the NEIL1 substrate relative to the baseline. This variant has two mutations in the substrate (num_mutations = 2). This feature has a SHAP score of −0.05 (blue), indicating that a higher number of

mutations in this RNA is unfavorable to editing. There is no A:C mismatch at the editing site, and this feature value has a SHAP score of −0.06. These and other highlighted feature values serve to drive the prediction down from the baseline of 0.25–0 (Fig. 7f).

To illustrate the directionality of predictive association of the features with RNA editing levels, we plotted the SHAP scores of the top 20 features for all test set variants of the three substrates (Fig. 7g). We also summarized the relative importance of features for each of the three substrates by computing the percent contribution from each feature to the mean of absolute SHAP values across all examples in the test sets of each substrate, and highlighted the six new features unique to this study in red ("probability_active_conf", "sim_nor_score", "ensemble diversity", "mfe_frequency", "site_5prm_cp_internal:C:G", and "d2_5prm_cp_internal:G:C", Fig. 8a). The closing pairs for loops and bulges are previously unexplored, but highly ranked features for NEIL1 and TTYH2 (Fig. 7g). Closing pair can be a readout for alternative active structure, such as in some highly edited NEIL1 variants the d2 internal loop's 5′ closing pair is a G:C sequence ("d2_5prm_cp_internal:G:C"; Supplementary Fig. 7) compared to the U:A pair in WT (Fig. 7b). These results also corroborate with trends observed in the abovementioned clustering analysis (Fig. 6, and Supplementary Figs. 7–9).

Eight features were illuminated as most important for driving model predictions across substrates. Number of mutations ("num_mutations") was the strongest contributor for AJUBA (47.69%) and NEIL1 (19.18%), and in the top six most important features for TTYH2 (4.51%; Fig. 8a). Increasing number of mutations had a negative influence on editing levels (Fig. 7g). This effect supports the proposal that RNA structure plays a big role in editing activity because in our library design the more mutations (single versus double-transversion) the larger changes occur in the structure (Fig. 4c). An A:C mismatch at the editing site ("site_1_1:A:C") had a high relative contribution for NEIL1 (18.03%) and TTYH2 (15.95%), but contributed less to AJUBA editing levels (0.31%), consistent with previous proposals that A: C mismatch facilitates the flip-out of the adenosine for ADAR editing[26,27] (Fig. 7g and Supplementary Fig. 4g). The probability of the active conformation ("probability_active_conf") accounted for a mean of 8.8% relative contribution across substrates. The structure-similarity score of variants compared to WT feature ("sim_nor_score") was positively correlated with editing levels. The lower MFE is positively associated with editing for NEIL1 and TTYH2, but not for AJUBA. The higher ensemble diversity is positively associated with editing levels for TTYH2 and AJBUA

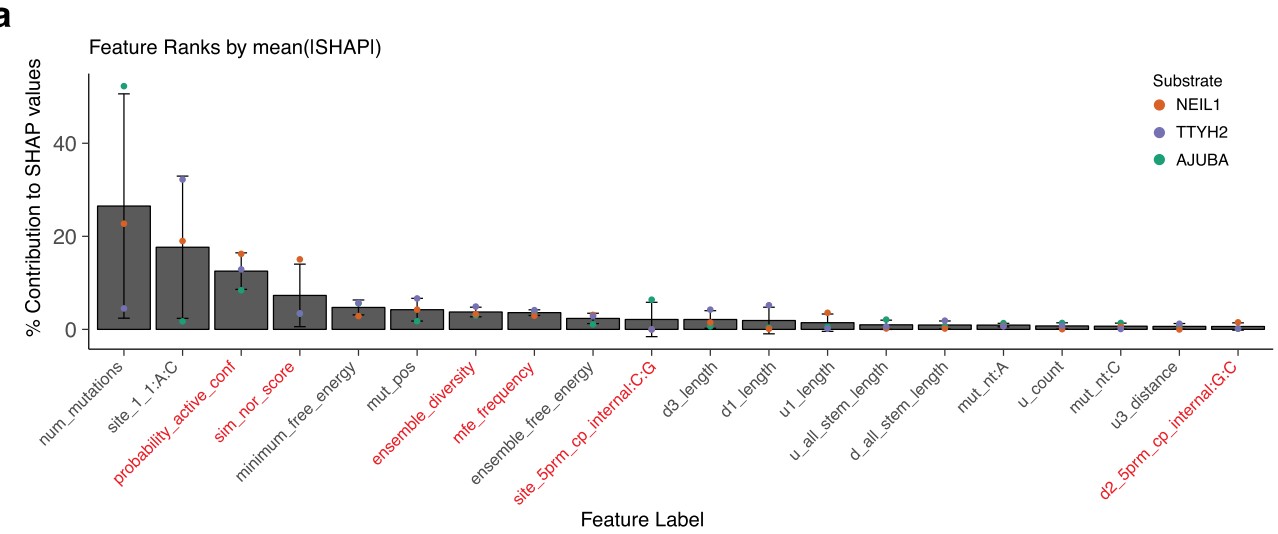

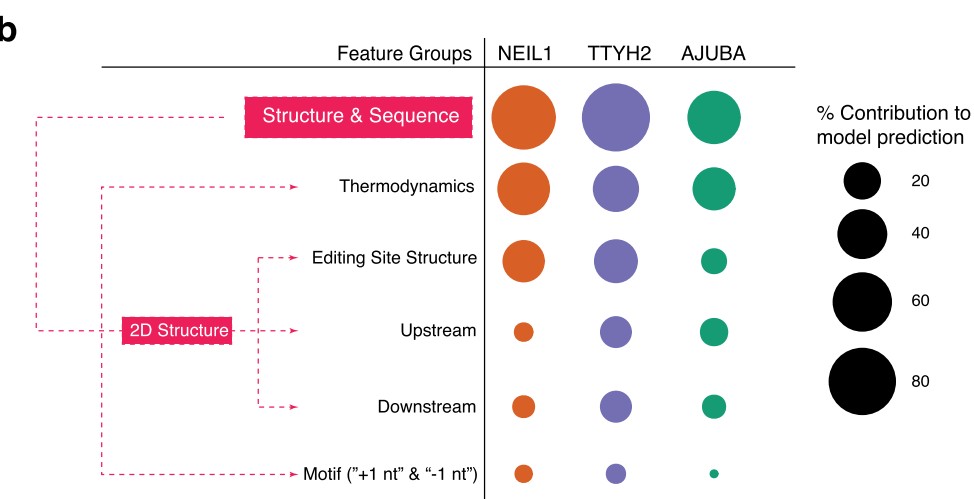

**Fig. 8 Cis-regulatory features synergistically contribute to model prediction. a** Percent contribution of individual feature to model prediction ranked by averaging normalized SHAP values. Error bars indicate the variability in feature contribution across the three substrates NEIL1 (orange dot), TTYH2 (purple dot), and AJUBA (green dot). The new features unique to this work is highlighted in red. Higher ranking with smaller standard errors indicates that these features are commonly among the highest contributors to model prediction in all three RNAs. **b** Contributions of different feature groups to the prediction of editing levels for each RNA library. NEIL1 (orange), TTYH2 (purple), and AJUBA (green). Black dots indicate the scale. The subgroups of individual features included in each feature group are listed in Supplementary Data 1.

but bidirectional for NEIL1. Higher MFE frequency is positively associated with editing levels for NEIL1 and TTYH2, but not for AJUBA (Fig. 7g). These features corroborate with previous results that the overall structural stability of RNA substrate is positively correlated with the editing activity[15] and reveal that RNA conformational diversity plays important and specific roles in different substrates. The seventh ranking feature was the position of the mutation along the RNA molecule ("mut_pos"). "Mut_pos" values are numbered beginning at the 5′ end of the RNA molecule, so higher values indicate positions further from 5′ and closer to 3′. This result indicates that the nucleotides adjacent to the editing site in the structure is the hot spot dictating activity. Though the "mut_pos" feature had a strong impact on editing level, the directionality varied across substrates, reflecting the interplay of the mutation position with other structural features.

In addition to top individual features, a sparse set of features collectively contribute to the accurate predictions made by the models. For the NEIL1 substrate, 90% of the explained variance could be attributed to the 26 top features, compared with the 32

top features for TTYH2 and 23 top features for AJUBA (Supplementary Data 8). To illustrate the contributions of different types of features and to draw biological insights, we looked at feature groups and subgroups. We categorized the group of all structure and sequence features excluding mutation-related features to four subgroups (Fig. 6b, full list of feature groups and subgroups in Supplementary Data 1). Overall, the thermodynamics and the editing site structure have the largest contributions, consistent with prior proposals that the overall thermodynamics (RNA stability and conformational diversity), and the structure of the editing site dictate the editing efficiency[11,15]. Notably, upstream and downstream structure features are also important, such as the downstream features in TTYH2. The −1 and +1 nt sequence motif (the 5′ and 3′ nearest neighbor, termed "site_prev_nt" and "site_next_nt" in Fig. 7f and Supplementary Data 1) also contributes to the prediction albeit to a lesser extent.

This systematic interpretation of our models reveals not only several biologically relevant features that are globally predictive

across the three substrates, but also some that are highly predictive for specific substrates. These results showcase the promise of predictive cis-regulatory models of RNA editing, but also highlight the need for much larger datasets spanning diverse substrates to learn more generalizable models of RNA editing.

## Discussion

The ultimate goal in understanding the cis-regulation of RNA editing is to develop a model that accurately predicts the ADAR editing efficiency in vivo, namely an "editing code". Unlike protein–DNA or protein–ssRNA interactions, where the primary cis-sequence largely dictates the interaction, ADAR substrates are required to bear double-stranded secondary structure. The difficulty of associating RNA sequence and secondary structure features to editing activity is a major challenge in studying the cis-regulation of RNA editing. To tackle this challenge, we integrated high-throughput measurements of ADAR editing with computational analysis of >100 RNA sequence and structure features simultaneously. The CRISPR/Cas9 engineering allowed us to study the cis-regulation of RNA editing by introducing desired mutations at the endogenous locus with the minimal perturbation of the RNA editing process. Our key results can be summarized in four main points. First, we found alternative structures that can be equally or better edited than the WT structure (Fig. 6, and Supplementary Figs. 7–9). Second, our models confirmed all known features identified by previous biochemical and transcriptomic studies (A:C mismatch at the editing site, the 5′ and 3′ nearest neighbors, the length and stability of the substrate including 5′ stem length and 3′ loop structure)[11–14,17] and revealed previously unexplored features, including ensemble diversity and closing pairs (Figs. 7 and 8, and Supplementary Data 8). Third, our substrate-specific machine learning models integrated diverse sequence and structural features to quantitatively predict editing levels of new variants for a given target (Fig. 7d). Fourth, both general and substrate-specific features synergistically contribute to editing levels, and the degree of contribution of each feature varies across different RNAs, suggesting complex and context-dependent cis-regulation of the RNA editing landscape (Fig. 7e–g). A lot of progress has been made in recent years in deciphering the RNA splicing code[30–32]. However, we just began to uncover the ADAR-mediated RNA editing code which harbors complex regulation via RNA secondary structure.

Our approach opens several new lines of inquiry for further improvement. Measuring the structure of RNA variants in cells[33–35] at native endogenous loci (versus relying on predicted structure) would greatly enhance the RNA structure analysis. In addition, advanced experimental methods such as irCLASH[36] may be applied together with gene-specific amplification to validate the ECS sequence. Further, while we focused on cis-elements adjacent to the editing site, long-range interactions, such as the editing inducer elements[37] important for editing can be investigated using our approach. Although our data focused on the editing level, which is largely determined by the deaminase domain[14,38], the dsRBD (illustrated in Fig. 5c) also contributes to substrate recognition[39,40]. In the future, high-throughput in vitro RNA-binding experiments can be performed to combine with existing SELEX data for dsRBDs[41] to identify features specific to dsRBDs of human ADARs using our pipeline. To tease apart trans-regulation effects by RNA-binding proteins (RBP)[42], in vitro competitive binding experiments and the editing measurements in cells can be conducted in the knockout or overexpression background for the RBP of interest. Because we observed no correlation of RNA abundance on editing level (Supplementary Fig. 2) and our experiment measures

pre-mRNAs, our editing analysis is likely not affected by potential effects of sequence variation on RNA processing. Nevertheless, understanding the interplay between RNA editing and various RNA biology, such as RNA processing pathways and RBP presents an important question for future investigations.

Several systems were recently developed to recruit ADAR enzymes to specific sites for site-directed RNA editing[43–49], providing novel tools to study biological function, and a safer and reversible alternative to gene therapy[43,50–52]. Currently, these RNA engineering methods mainly use antisense gRNAs that form perfect duplex with the target region except for an A:C mismatch at the editing site. Our results strongly support additional imperfectly base-paired designs to mimic the highly selective and efficient editing observed in the natural ADAR substrates with complex structure features. Such features include relatively short 5′ (upstream) stem required for ADAR1 (ref. [14]) compared to ADAR2 and specific non-stem 3′ (downstream) structure, where the internal loops at the 3′ likely contribute to the ADAR selectivity[17]. Notably, each of the three RNA substrates we tested has substrate-specific features that dictate the editing efficiency (Fig. 7g). This showcases that a screen of possible designs of gRNA would be a valuable and cost-effective strategy to learn the best features that lead to the most specific and efficient editing for each different target site in transcriptome engineering. In this regard, our experimental methods and computational pipeline are readily applicable.

There are several limitations to the modeling approaches utilized in this study. While our current results give rise to models with substantial predictive power for individual substrates, their generalizability remains low (Supplementary Fig. 11). This tendency to overfit will be mitigated by expanding the training set to include more RNA substrates, allowing the model to learn the shared properties of RNA substrates. Furthermore, the SHAP interpretation of feature importance in the XGBoost model highlighted the significance of features related to mutation number, structure, sequence, and position. This result suggests that an effective featurization of the data relies upon knowledge of a WT substrate structure and sequence, which may not be available for all substrates. Nonetheless, any variant in a library can be arbitrarily assigned as the WT for featurization purpose.

Building on our work using the PREUSS pipeline, ADAR editing can be further investigated in larger scale and in different cell types, tissues and disease states to explore the full spectrum of cis-regulation. Ultimately, establishing the "RNA editing code" will help us better understand the underlying rules of RNA editing, and facilitate efficient and precise transcriptome engineering for studying RNA biology and treating human disease.

## Methods

**Cell culture and transfection**. HEK293T cells (ATCC) were cultured in Dulbecco's modified Eagle medium (Life Technologies) supplemented with 10% FBS (Gibco, Thermo Fisher) and penicillin streptomycin (Life Technologies). Cells were maintained at 70–90% confluency. One day before transfection, ~700,000 cells were split to 6-well plates. The next day, 500 ng of Cas9–sgRNA construct in the px330 backbone (https://www.addgene.org/42230/) was co-transfected with 500 ng of the DNA donor using lipofectamine 2000 (Invitrogen). Cells were maintained at 50–90% confluency for 5 days.

**Design of the CRISPR/KI donor oligos**. We selected three natural ADAR1 substrates (NEIL1, TTYH2, and AJUBA; Fig. 1b and Supplementary Data 7) for the mutagenesis studies based on the observations from available RNA-seq data that (1) the editing sites for all three substrates are highly edited (30–60%) in HEK293T cells, in which ADAR1 is expressed but ADAR2 is lowly expressed; (2) the editing sites are not edited when ADAR1 activity is abolished; and (3) they represent three different types of dsRNA substrates. The NEIL1 editing site is in the coding region. The editing event leads to an amino acid change from lysine (K) to arginine (R), which has been shown to increase the enzymatic activity of the NEIL1 glycosylase[53]. The TTYH2 editing site is intronic and the AJUBA editing site is

located in its 3′ UTR. The functional impact of these two editing sites is currently unknown.

Two types of CRISPR KI donors were designed in this study: the degenerate donor and the fixed donor. For the degenerate donor oligos, a single-stranded DNA oligo was synthesized in which degenerate sequences were introduced at the interrogated regions. In the NEIL1 donor (Fig. 1c), −3 to −1 and +1 to +3 were interrogated and equal molar of four nucleotides were introduced at these positions during DNA synthesis. To avoid cutting by the Cas9, a point mutation was also introduced at the PAM sequence. In the TTYH2 donor (Supplementary Fig. 1a), a 10 nt region in the ECS was studied, and equal molar of C or T was introduced. The PAM sequence was also mutated along with a compensatory mutation to maintain the secondary structure.

For donors used for targeted mutagenesis, individual DNA sequences were designed to carry desired mutation(s). Briefly, a 15–20 nt region around the target editing site and the corresponding region on the opposite strand were subject to mutagenesis. All possible nucleotide at any single position was tested, with exceptions where A-to-G mutation was avoided in the +1 and −1 nt of NEIL1 because it potentially becomes indistinguishable with A-to-I editing in RNA-seq results. Combination mutations at two positions were also designed, in which each of the positions is mutated to the nucleotide in the opposite strand to disrupt the original structure. In addition, individual donors with altered length for interrogation of specific features of the RNA substrate were included. For NEIL1, we were able to use donor oligos to introduce compensatory mutation variants because the ECS and editing site are close in sequence space. All oligo sequences used in this study are listed in Supplementary Data 6.

**Generation of the CRISPR/KI donor pool**. For NEIL1 donors, 80mer oligos were purchased from IDT and pooled at equal molar ratio. Oligo pairs NEIL1_leftarm/ NEIL1_rightarm, or asymmetrical labeled primer pairs NEIL1_leftarm_biotin/ NEIL1_rightarm and NEIL1_leftarm/NEIL1_rightarm_biotin (Supplementary Data 6) were used separately to add additional sequences to obtain 200mers in PCR reactions using Phusion polymerase. Around 400 μl PCR products were purified using MinElute PCR purification kit (Qiagen) to obtain the dsDNA donor pool, which was verified by agarose gel electrophoresis. For single-stranded donors, 100 μl MyOne Streptavidin Dynabeads (Thermo Fisher) were added to the purified products that were amplified with asymmetrical biotin label, and then the mixtures were denatured at 95 °C for 10 min and chilled on ice immediately. The unbound single-stranded oligos were collected from the supernatant and then purified with column MinElute PCR Purification Kit (Qiagen) to obtain the ssDNA donors. For TTYH2 and AJUBA donors, 100–120mer pooled oligos were purchased from Agilent and amplified using individual primers (Supplementary Data 6). Primers donor_F and donor_R of each target gene were used to specifically amplify the oligo library from the oligo chip. A second PCR using Donor_F_70 and Donor_R_70 was performed to elongate the homologous arms of each donor. The PCR products were purified using MinElute PCR purification kit (Qiagen) and used as dsDNA donors later. NEIL1_degenerate_donor and TTYH2_degenerate donors were synthesized as Ultramer by IDT and used directly in the transfection.

**Guide RNA design and cloning**. gRNA was predicted by the web-based software CRISPR.mit.edu. The higher ranked gRNA with a PAM sequence close to the interrogation region was selected. For the TTYH2 and AJUBA loci, different sets of gRNAs were designed for the KI regions in two opposite strands. To construct gRNA plasmids, two reverse complementary single-stranded oligos with overhangs were synthesized by IDT and annealed on a thermocycler (Bio-Rad) before ligation to BbsI-linearized PX330 backbone. The ligation mix was transformed into Stbl3 chemical competent cells (Invitrogen) and single clones were sequence verified by Sanger sequencing.

**CRISPR mutagenesis and library construction**. We used 600 ng single-stranded oligo donor library or 1200 ng double-stranded oligo donor library along with 500 ng gRNA construct to co-transfect into 1 million HEK293T cells using lipofecta-mine 2000 (Invitrogen). For degenerate donor mediated KI, 1 μl of 10 μM degenerate donor was used. A total of 1 μM L755507 (Sigma Aldrich) was added to the media 1 day after transfection to enhance the HDR efficiency[54]. Two biological replicates were included for each assay. The transfected cells were grown for 5 days before they were seeded onto 10 cm dishes for an additional two days. A total of 10% of the cells were harvested for genomic DNA using Quick-DNA kits (Zymo Research). The remaining cells were used for nuclear extraction using the Nuclear/ Cytosolic Fractionation Kit (Cell Biolabs) following the manual. Nuclear RNA was purified from the nuclear extract using the Trizol method. Genomic DNA was removed from the RNA samples using the TURBO DNase (Thermo Fisher Sci-entific). The primers were designed to make sure pre-mRNA species were amplified for RNA editing analysis. RT was performed using SuperScript III kit (Thermo Fisher Scientific) and the gene-specific primers. All RT products were used in total of 300 μl (50 μl × 6) PCR reaction with Phusion polymerase (Thermo Fisher Sci-entific), and gene-specific primers with Fluidigm mmPCR adaptor sequences[55]. Genomic DNA library was amplified using a similar approach, except for the different primer set. All first round PCR products were size-selected on 1.5% agarose gel and purified using Gel purification Kit (Qiagen). Diluted PCR product

(1:50) was used in the second round of PCR to add the Illumina sequencing adapter and individual barcode sequences, using Fludigm_universal_F/flu-digm_barcode_R (ref. [55]). The library was size selected and purified as in the previous step.

**Next-generation sequencing and data analysis**. All libraries were sequenced on a NextSeq550 using Basespace (Illumina) for data collection. NEIL1 libraries were sequenced for 75 cycles paired-end and TTYH2 and AJUBA libraries 150 cycles paired end. Quality of reads were evaluated, and reads were filtered by FastQC default settings. To map the variants of the target gene, a reference genome was first built using the GMAP package, where designed mutations were included as SNPs. Briefly, GSNAP was used to detect variants with mismatches inside the interrogated region but not indels. The mapped reads were separated into indi-vidual variants based on the unique mutations carried in the region except for the editing site, and RNA editing was called and measured for each variant, as described previously[56]. The indel variants were mapped individually. The editing level Z-score of each variant is calculated for each RNA library by Eq. (1) as:

$$Z_i = \frac{EL_i - EL_{WT}}{S} \qquad (1)$$

where $S = \sqrt{\frac{\sum_{i=1}^{N}(x_i - \bar{x})^2}{N}}$; EL = editing level; $x_i = EL_i - EL_{WT}$; $\bar{x}$ = mean editing level for a given library.

**Chemical mapping of RNA structure in vitro**. We were able to construct RNA libraries by in vitro transcription by T7 polymerase (Megascript kit, Thermo Fisher) for the NEIL1 and a portion of TTYH2 (TTYH2-ECS library) variants to probe th e RNA structures in vitro to compare with computationally predicted structures (below). The NEIL1 library (DNA oligo manufactured by IDT) was constructed with 3′ common primer binding sequence (PBS) and 3′ hairpin bar-codes similar to previous report[57] (see Supplementary Data 6). For the TTYH2-ECS library (oligo manufactured by Agilent), we designed new 5′ PBS and 3′ barcodes as listed in Supplementary Data 6. The DMS and ethanol-control experiments were performed according to reported protocols[57] except for the reverse transcription step was carried out using the TGIRT-III enzyme (Ingex), which improved efficiency of the reverse transcription reaction (50 mM Tris-HCl pH8, 75 mM KCl, 3 mM MgCl₂, 5 mM DTT, 1 mM dNTPs, 100 U TGIRT-III enzyme, and 10 U SuperaseIN)[35]. The reverse transcription reaction mix (12 μl) were incubated at room temperature for 5 min prior to incubation at 57 °C for 3 h followed by quenching of reaction by adding 5 μl of 0.4 M NaOH at 90 °C for 3 min and then cooled on ice for 3 min by adding 5 μl acid quench mixture (1.43 M NaCl, 0.57 M HCl, and 1.29 M sodium acetate pH 5.2). The first strand cDNA was then purified by RNAclean XP beads and amplified by one round of PCR to construct the library to add index and barcodes (Supplementary Data 6). The resulting library was sequenced with pools of diverse sequences to increase read quality by NextSeq550. NEIL1 library was sequenced by paired-end on 2 × 76 cycles and TTYH2 on 2 × 150 cycles. Sequencing data were collected by Basespace (Illumina). Reads were first filtered by AfterQC[58] ("-q 30 -f0 -t0", quality threshold, "30", no trim on both ends) then mapped and demultiplexed by cutadapt 1.17 to read and trim the barcodes in three steps[59] (first remove common sequence, −e 0.07, resulting error rate, −7%; second detect the barcodes, −e 0.15, error rate, 15%; third by detecting "common sequence + barcode" in wildcard mode in the rest of unrecognized reads, −O 36, minimum overlap, −36). The resulting reads were processed by ShapeMapper 2 (ref. [60]; default configuration except for read depth threshold was set to 2000) to detect DMS reactivity followed by structure inferring by Biers in MATLAB[61] (default settings except for max_bootstrap = 100). For NEIL1, two replicates are performed by conducting the chemical treatment on separate tubes, while the replicates of TTYH2-ECS are from two different barcodes of the same RNA in one experiment. DMS reactivity data and experimentally inferred MFE structure were deposited in the RMDB database[62].

**Computational RNA secondary structure prediction**. The sequence used for WT NEIL1, TTYH2, and AJUBA are shown in Supplementary Data 7 and Fig. 1b. We chose the ECS sequence according to the reported method[18] by examining the 100–1000 bp sequence flanking the editing site and choosing the most sData RNA duplex. For AJUBA, the ECS sequence we predicted also matches the predicted duplex using the RNAhybrid method[63]. We chose the region flanking the editing site of AJUBA to fold a "minimum" hairpin structure as the AJUBA RNA substrate (Fig. 1b) by omitting a 524 nt sequences in lieu of the full length (>800 bp). This is because this minimum AJUBA RNA structure preserved both the base-pairing and a natural loop structure (shown as the hairpin loop in Fig. 1b) based on the resulting duplex structure, using the ECS prediction results mention above. The secondary structures with the MFE of the RNA variants (Supplementary Data 7) for all three RNAs are calculated from the Vienna RNAfold[64] 2.4.14, using default parameters except for allowing lone pairs (parameter: -p -d2).

To evaluate if the computationally predicted MFE structure is similar to the RNA structure probed experimentally in vitro, we used the SimTree[65] method version 1.2.3 to compare the MFE structure between computationally predicted to the experimentally inferred structures (described above) for all NEIL1 variants and a portion of the TTYH2 variants that we were able to experimentally measure. The

MFE structures are largely similar between experimental and computational results shown by the high average value of the pairwise normalized similarity score calculated using SimTree[65] (NEIL1 = 0.96 ± 0.08, TTYH2-ECS library = 0.97 ± 0.04, where 1 means identical structures, Supplementary Fig. 4c). We reasoned that the results from this comparison justify the suitability of using the computationally predicted structure in our analysis. Therefore, for all of the structural feature analysis, we used the computationally predicted RNA structures to be consistent for all three RNA libraries.

**Clustering analysis of RNA sequence and structure**. We performed clustering analysis for each RNA library using the LocARNA pipeline (version 2.0.0RC8)[23,24] and associated tools. First, we used both the RNA sequence and the MFE structure (Supplementary Data 7) as the input to the mlocarna module[23] to generate a multiple alignment. The resulting multiple alignment was then input into RNAclust[66] (RNAclust.pl, version 1.3, modified to suit current computing environment) to generate a hierarchical cluster tree file (Fig. 6a and Supplementary Fig. 8) and the consensus RNA structure for each cluster (Fig. 6b, and Supplementary Figs. 7b and 9). These hierarchical clustering and the consensus RNA structure for each cluster can also be viewed in SoupViewer[66]. The hierarchical clustering was illustrated (Fig. 6 and Supplementary Fig. 8) using dendrogram generated by the iTOL web server[67].

**Calculating the probability of forming wild-type secondary structure**. The probability of forming WT-like RNA secondary structure was calculated with Vienna RNAfold[64] version 2.1.9. The probability of forming the WT-like secondary structures was calculated by Eq. (2) as:

$$\text{Probability} = \frac{e^{\left(\frac{-E\_\text{wt}}{kT}\right)}}{Z} \tag{2}$$

where $kT = 0.6$ kcal at temperature $T = 37\,°C$. $Z$ is the unconstrained partition function (calculated with RNAfold -p). $E\_\text{wt}$ is the energy of the state with the WT-like secondary structure, calculated using both the constraints and reference information (listed below) in RNAfold. The details of the usage and the selection criteria for these constrains and reference information are also explained below.
The structure constraints are:
NEIL1:
……………………………………...>>>….>…>>…>…>…
TTYH2:
………………………………………………..>>>>>>>>>>.>.>.>>>>>>>>>….
AJUBA:
…………………………………………..>>.>>>>.>.>>>.>>>>.>>>>>>.

where ">" indicates that the given base must be paired with a residue that comes before it (5′) in the sequence and "." indicates no constraint for the given base.
The reference information is:
NEIL1
………………..(((((.(((.(((((……))))).))).))))))…………………
TTYH2:
…………………((((((.((.((((((((((((…………………..))))))))))))).)).))))))…………………..
AJUBA:
((((((((((((((((.(((((((.(.(((((((((((.(((……….)))).)))))))))).)).))))))).)).))))))).)))))))))))))))).

The constraints are used to specifically calculate the free energy when fold the RNA sequence according to the constraints. The reference information was used to calculate a penalty by further comparing the base-pairing between the folded structure and the structure indicated by the reference information. We incrementally tested different versions of both the constraints and the reference information, starting from the exact secondary structure of WT to gradually relax the base-pairing starting from the first base-pair below the hairpin loop. This way the constrains and reference information together would preserve majority of the base-pairing information in WT near the core region of the RNA (regions that flanking the editing site). The reference information is used so that an additional penalty was added if base pairs could not be formed in the core region of the RNA. Note that penalties are not applied for any additional base pairs that form in the unpaired regions of the reference information. The probability was divided by the number of noncanonical base pairs that would be formed in the core of the WT secondary structure (defined by the reference information) to roughly account for the additional energetic penalty that these base pairs should incur. These constraints and reference information also ensure that the "active conformation" we calculated includes a wide group of core conformations that closely resembles the MFE structure of the WT, but are not limited to the single WT MFE structure. Additionally, the final versions we choose (listed above) are the ones that can compute conformations existed in the majority of RNA variants (88% for NEIL1, 96% for TTYH2 and AJUBA). The script in is available on GitHub (URL: https://github.com/kundajelab/PREUSS.)

**Modeling the 3D structure of ADAR1 bound to NEIL1**. A 3D model of ADAR1 bound to NEIL1 was built through homology modeling and the Rosetta RNP-denovo method[68]. First, a homology model of human ADAR1 deaminase was built using Phyre2 (ref. [69]). The conformation of the core RNA residues (nucleotides

corresponding to NEIL1 residues 30–39 and 44–53) was taken from the previously solved structure of human ADAR2 bound to double-stranded RNA (PDB ID: 5HP3). The RNA was positioned relative to the protein by aligning the previously solved ADAR2 structure (in complex with dsRNA) to the homology model of ADAR1, then copying the RNA coordinates from the ADAR2-dsRNA structure. Protein residues in the ADAR1 homology model that clashed with the RNA were removed (the final residues included in the model were: 823–973, 996–1003, and 1010–1223). This model was used as input to RNP-denovo with the -s option. Helical regions of the NEIL1 RNA were modeled as ideal A-form helices, also included with the -s option. Conformations of protein residues were not optimized (-minimize_protein_sc false and -rnp_high_res_cycles 0). Default settings were used for all other options. The complete RNP-denovo command line used is provided below:

rna_denovo -fasta fasta.txt -secstruct_file secstruct.txt -s ADAR1_homology_model_and_core_RNA_from_5hp3.pdb RNA_helix_1.pdb RNA_helix_2.pdb RNA_helix_3.pdb RNA_helix_4.pdb RNA_helix_5.pdb -new_fold_tree_initializer true -minimize_rna true -minimize_protein_sc false -out:file:silent build_full_wt_neil1.out -rna_protein_docking true -rnp_min_first false -rnp_pack_first false -cycles 10000 -rnp_high_res_cycles 0 -minimize_rounds 2 -nstruct 2000 -ignore_zero_occupancy false -convert_protein_CEN false -FA_low_res_rnp_scoring true -ramp_rnp_vdw true -dock_each_chunk_per_chain false -use_legacy_job_distributor true -no_filters
where RNA_helix_1.pdb, RNA_helix_2.pdb, etc. are ideal A-form helices for base-paired regions of Neil1. Possible placements of the double-stranded RNA-binding domains were visualized by aligning the previously solved structure of the ADAR2-dsRNA-binding motif bound to dsRNA (PDB ID: 2L2K) to our model of NEIL1 bound to ADAR1.

**Machine learning models of RNA editing levels**. All feature extraction and model training code are available to access on github: https://github.com/kundajelab/PREUSS

*Feature extraction*. RNA structures for NEIL1, AJUBA, and TTYH2 were annotated with the bpRNA algorithm[9]. The bpRNA annotations were in turn utilized to extract structural and positional features for each variant. A feature matrix with structure-specific features from the bpRNA (annotations, sequence-specific features, features that take into account the mutation type and position, and thermodynamic-specific features was engineered (how the featured were derived are described in Supplementary Data 1)) for each substrate and included a total of 122 features (Supplementary Datas 2–4).

*Model training*. The XGBoost[28] Python library (v. 0.81) was used to train gradient boosted regression trees to predict Adar editing levels from feature matrices described above. Training was performed both within-substrate and across substrates. Several training and test combinations of datasets were utilized and summarized in Supplementary Data 5.
The dataset was randomly separated into three splits: training on 70% of variants, model validation on 15%, and testing on the remaining 15%. To avoid train/test contamination, base-pair positions along the RNA molecules were assigned to one of the three splits (training, tuning, or test). All features associated with a given base pair position were assigned to the corresponding split. Any feature that was null or non-varying across all variants in a given training split was removed from analysis. A number of variants were characterized by two or more mutations relative to the WT. For these, features were defined for each mutated base pair separately. To avoid train/test contamination, any base pairs that were both mutated in a given variant were included in the same split. The rationale for defining features relative to bases rather than variants is that different combinations of mutations may lead to variations in editing level, and a number of features were derived in reference to mutation type and position (Supplementary Data 1).
XGBoost was trained for a maximum of 1000 iterations, with early stopping after ten subsequent rounds with no reduction in root mean square error (RMSE) on the validation split. Default parameters were used.
The $R^2$ value was calculated on the test set to determine the percent of total variance explained by the feature matrix. Other metrics to measure model performance included:

- Spearman correlation from the scipy.stats Python library.
- Pearson correlation the scipy.stats Python library.
- Mean absolute error (MAE) from sklearn.metrics Python library.
- Mean absolute percent error (MAPE).
- Root mean square error (RMSE) from sklearn.metrics Python library.
- Area under the precision-recall curve (auPRC) from sklearn.metrics Python library.
- Area under the receiver operating characteristic (auROC) from sklearn.metrics Python library.

*Feature importance analysis*. Feature importance analysis was performed to identify the subset of features most informative in predicting Adar editing levels. The XGBoost "plot_importance" function was used to calculate the $F$ score for each

feature. The TreeSHAP algorithm[29] was applied to interpret feature importance from the XGBoost model. SHAP summary values were computed for each feature as a measure of feature importance using the "shap_values" function within the "TreeExplainer" class. Pairwise interaction values from TreeShap were also calculated to identify highly correlated feature values.

SHAP values were applied to calculate the combined relative importance of feature subsets. Feature subsets (Supplementary Data 1) were defined as follows; some features were parts of multiple subsets:

- Structure features: stem length, free energy, probability of active conformation.
- Number of mutations in the variant.
- Mutation-specific sequence features: mutation position, mutation site reference allele, mutation site alternate allele, distance of mutation site from edited base.
- Mutation-specific structure features: bpRNA structure designation for the mutation site, bpRNA structure designation for the adjacent upstream site, bpRNA structure designation for the adjacent downstream site, boolean indication of whether or not the mutation is part of the same structure as the editing site.
- "Other" mutation-specific features: type of mutation (indel, SNP), presence/absence of mutation (WT/mutated) in the variant.
- Editing site sequence features.
- Editing site structure features.
- Characterization of the 3 bpRNA structural features upstream of the editing site.
- Characterization of the 3 bpRNA structural features downstream of the editing site.

For each feature subset, the mean absolute SHAP values across variants were calculated. These were in turn summed across all features in the subset and compared to the total sum of mean absolute SHAP values across all features.

Overall feature rankings were computed by calculating the mean absolute value of SHAP values for each feature across the test set samples. These mean(|SHAP|) values were summed across all features, and the percent contribution to the total was obtained for each feature. These percent contributions for each feature were averaged across substrates to determine features that were ranked as high importance consistently across all substrates.

**Reporting summary**. Further information on research design is available in the Nature Research Reporting Summary linked to this article.

## Data availability

The data supporting the findings of this study are available from the corresponding authors upon reasonable request. Data related to all figures are provided in the Supplementary Data files 1–8. The accession number of the RNA-seq data for measuring RNA editing is GSE138860. For the DMS chemical probing of in vitro RNA structure, raw data and ShapeMapper 2 processed data are available at GEO database with accession number GSE168234, and the DMS reactivity data and inferred RNA secondary structure are available in the RNA Mapping Database (RMDB), the RMDB IDs are: NEIL1_DMS_0001, NEIL1_DMS_0002, NEIL1_DMS_0003, NEIL1_DMS_0004, NEIL1_DMS_0004, NEIL1_DMS_0006, NEIL1_DMS_0007, NEIL1_DMS_0008, NEIL1_DMS_0009, NEIL1_DMS_0010, NEIL1_DMS_0011, NEIL1_DMS_0012, NEIL1_DMS_0013, NEIL1_DMS_0014, NEIL1_DMS_0015, NEIL1_DMS_0016, NEIL1_DMS_0017, NEIL1_DMS_0018, NEIL1_DMS_0019, NEIL1_DMS_0020, NEIL1_DMS_0021, and TTYH2_DMS_0001.

## Code availability

Bioinformatics codes for RNA editing call are available upon request. Codes for the PREUSS computational pipeline is available on GitHub URL: https://github.com/kundajelab/PREUSS (https://doi.org/10.5281/zenodo.4563064)[70].

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

## Acknowledgements

We thank members of Li, Kundaje, and Das laboratories, especially Dr. Patricia Deng, Dr. Anne L. Sapiro, Dr. Shibin Hu, and Dr. Avanti Shrikumar for insightful discussions on the machine learning analysis, and Wipapat Kladwang and Dr. Joseph D. Yesselman for suggestions on RNA structure analysis. We thank Lei Shi as a computational consultant. This work is supported by National Institutes of Health (NIH; GM124215 and GM102484 to J.B.L.), the Milton Safenowitz Postdoctoral Fellowship from the ALS Association (to T.S.), and the Stanford Bio-X Bowes Fellowship (to A.S.).

## Author contributions

X.L., T.S., G.R., and J.B.L. conceived the work. X.L., A.S., T.S., I.J., A.K., J.B.L., and R.D. co-wrote the manuscript. T.S. designed and carried out the CRISPR/Cas9 and RNA editing measurements. T.S. and Q.L. carried out the editing level analysis. X.L. carried out the RNA clustering analysis, and performed the RNA chemical mapping experiments and data analysis. X.L. and K.K. performed RNA and protein structure analysis. A.S., X.L., and A.K. developed the PREUSS computational pipeline.

## Competing interests

The authors declare the following competing interests: J.B.L. is a co-founder of AIRNA Bio and a consultant for Risen Pharma. Anna Shcherbina receives consulting fees from Myokardia, Inc, is a scientific adviser to Ravel Bio, Inc., and an employee of Insitro, Inc.
