## [Peer Review File · Nature Communications]

Reviewers' comments:

Reviewer #1 (Remarks to the Author):

Here the authors describe a screening approach for studying the effect of sequence changes in ADAR substrates with the goal of understanding (and predicting) cis-regulatory elements present in the RNA. With a clever CRISPR-based mutagenesis approach they introduce a large number of nucleotide changes at the genomic loci for three different ADAR substrates and use NGS to evaluate the effects these changes have on editing efficiency. They go on to employ machine learning to define the relationship between the various RNA sequence and structural features altered and the editing efficiency observed. They find that their models are reasonably effective at predicting editing levels within a group of closely related sequences, but are not effective at predicting editing levels for substrates that differ in sequence substantially from the training set.

ADARs have received a lot of attention recently given their role in human disease and their use in directed editing applications. Understanding how substrate structure controls editing efficiency is a long standing problem in ADAR research. While certain features of an RNA substrate (e.g. substantial duplex structure, A-C mismatch, 5' and 3' nearest neighbor preference, etc) have been known for some time to be important for editing, it is not currently possible to accurately predict editing efficiency from RNA sequence and structure alone. Therefore, the topic of this paper is very important. I do question the impact of the study in its current form, however.

Major comments:

1) The study involves the introduction of mutations in the genomic loci for three ADAR substrates and editing takes place with endogenous expression levels of these substrates and the ADARs (likely ADAR1 in HEK293T cells). All changes in editing levels are interpreted to arise from changes in the ADAR-RNA interaction. However, minimal consideration is given to how the introduced mutations may have altered other types of RNA processing events that will occur in these cells. For instance, do the sequence changes made to the NEIL1 substrate have any effect on splicing? Changes in splicing could have an indirect effect on editing efficiency. Or...do the sequence changes made in the AJUBA substrate have any effect on the transcript half life given its location in the 3' UTR? Sequence changes could also alter the interaction with competing RNA-binding proteins. Is it possible that sequence-dependent effects on the cellular metabolism of these RNAs are complicating the analysis and limiting the broad application of the predictive models? These issues should be addressed (or at least discussed).

2) The authors state in the abstract that this work "provides guidance for designing and screening of antisense RNA sequences that form dsRNA...for ADAR-mediated transcriptome engineering". However, it is not clear from the discussion how this is true other than this study suggests that predicting important features doesn't really work very well for new sites so one should screen for effective guide strands instead of trying to use prediction. Can the authors use the tools they have generated to predict effective guide strand designs for a therapeutically relevant directed editing target site? If so, they should provide an example. If not, then the "provides guidance for designing" statement should be removed from the abstract.

3) The authors state in the discussion that their results "provide new possible designs to mimic the highly selective and efficient editing observed in some natural ADAR substrates". However, the examples they give for "new" designs are features that had already been shown to be important for ADAR reactivity. The authors should clearly describe any truly new features they have identified as important and general (not just useful for controlling editing on one substrate).

Minor comments:

- 1) Top of page 5: "pyridine" should be "pyrimidine"
- 2) Page 8, Figure 5d should be Figure 5e?
- 3) Figure 5e should be Figure 5f?
- 4) Fig. 5f should be Fig. 5g?

Reviewer #2 (Remarks to the Author):

Using a CRISPR/Cas9 strategy, this manuscript presents new data on the effect on editing rates of mutations to three human ADAR1 editing sites with the overall goal of better understanding ADAR1 targeting. Specifically, they measure the impact of some sequence and secondary structural features with CRISPR/Cas9-mediated mutagenesis and use a machine learning model to predict substrate-specific RNA editing levels and further investigate contributions of a set of 122 features. This model, PREUSS, when trained on data on some of these mutations, is able to predict with high accuracy, the impact of other mutations.

Better understanding of ADAR targeting would permit better design of antisense oligos to target specific RNAs for editing via endogenous ADAR pathways, potentially opening a new strategy for designing targeted gene therapies. Also, ADAR is a double-stranded RNA-binding protein, more insight into its target preferences would also contribute to an overall understanding of how this important class of proteins identifies their target sites.

There has been some effort detailing sequence and structural preferences of ADAR targeting, this is the first attempt that I know of that comprehensively queries the impact of a large number of targeting mutations on editing rates at multiple targets. These data have the potential to inform detailed models of ADAR targeting that generalize beyond the three sites studied here.

Unfortunately, this manuscript does not achieve this goal. Specifically, the manuscript demonstrates, convincingly, that while it is possible to predict the impact of mutations to a target site that the classifier is trained on, these predictors do not generalize to other target sites. As such, it is not clear to me that the manuscript achieves any of the claims in the abstract and introduction. Specifically, it is not clear what additional new insight it provides about the general ADAR editing code (above what was already known and referenced in the manuscript) – the general editing code features shared across the three sites were already known, or seem vaguely defined. Nor it is clear what new guidance it provides for designing antisense RNA sequences for transcriptome engineering: if PREUSS is editing site specific, how will it help design antisense oligos for target sites it was not trained on? Finally, there seems to be little to recommend this modelling strategy as a foundation for developing predictive models of RNA editing – the design of the PREUSS features depend heavily on the WT site, what is the WT site for new sites for transcriptome engineering? What is the WT site when being used to detect new editing sites generated by SNPs or somatic mutations? Also, there is an additional question of novelty of the modelling strategy compared to recent work on predicting efficacy of CRISPR guides (PMID: 26780180, PMID: 29998038, PMID: 28263296), this work is not cited in the manuscript and seems to be the most similar work, rather than the splicing code work that is cited.

In addition to these concerns, it appears that the manuscript includes an incorrect, or old, draft of the computational methods section. There are a number of editing errors and incorrect citations in that section, some of which I have detailed below. This section requires extensive revision before it is suitable for review.

Detailed comments:

1. The authors mentioned that the CRISPR oligos used in the study include all possible single mutants and 11% of all possible double mutants for NEIL1. How were these double mutants selected? The authors provided a heatmap of editing levels of double mutations in the editing strand of NEIL1 on figure 2d, and concluded that, for double mutants, at least one mutation close to editing site would result in stronger effects. Given that only 11% double mutants were tested and they are all manually selected, are those samples representative enough to generalize about the feature effects for double mutants?

2. For secondary structure prediction and mutation type annotation of AJUBA, the manuscript uses a dsRNA stem formed by a complementary sequence 500+nt away from the editing sequence. Is this the verified ECS for the editing site? How did the author choose this ECS?
3. In particular, the AJUBA site seems particularly problematic. Perhaps some of the unique properties of mutations to the editing site are because there are other potential ECSs within the 500nt gap besides the proposed one, so mutations to the editing site will influence the choice of ECSs. The folding strategy for the AJUBA site is problematic too because it requires inventing a loop sequence which does not exist. Perhaps a tool like RNAhybrid would be more appropriate?
4. ECS is an awkward abbreviation especially because "editing site" is not abbreviated
5. Why is it important to choose target sites in different transcript regions? This was not clear to me in the motivation.
6. It seems strange that three mutations abolish editing in all cases, but some cases of 4 and 5 mutations do not. Why do you think this is?
7. What do you mean by "using dsDNA as a donor" (main text, near reference to Supp Fig 1j-k?)
8. The notation "transversion" and "transversion+break" is confusing. Why wouldn't a transversion always cause a break? It seems like what is meant by transversion is that the base altered is in a loop. If so, the notation could be clearer here.
9. The claim that "primary sequence might have a larger influence on editing of the AJUBA RNA" seems unsupported by the data. In particular, as I mentioned above, because it is such a long range interaction, there could be an alternative ECS which is favored by the transitions.
10. Also, it is not clear how to judge whether a difference between -71 kcal/mol and -87 kcal/mol is meaningful. Presumably, if both free energies are high enough to ensure stable dsRNA formation, there is no real functional difference between them.
11. The claim: "these observations indicate that RNA primary sequence and structure are intertwined in terms of mediating effects of mutations..." seems vague. What would be the alternative be? Isn't this what you would have already expected given the prior work on ADAR targeting? Furthermore, in the next paragraph, there is a further claim that "mutations may need to impact secondary structure to have major effects on RNA editing efficiency" which seems to invalidate the previous claim that there is an interplay.
12. The fact that number of mutations from the WT site is the most predicted feature is particularly disappointing as it doesn't really provide insight into the editing code nor permit generalization to new sites. Why was this feature used? What happens if it is not included?
13. The computational methods section needs editing. First of all, it is incomplete and details needed to reconstruct the methods are missing. Also, the citations are incorrect: e.g. the bpRNA is not described in citation 30. Finally, it is repetitive: we are told twice in the same paragraph that a temperature of 37C was used with RNAfold. The constraint description for the three target sites do not seem correct – why would one force the 5' and 3' stem flanking regions to be unpaired for NEIL1, TTYH2? Further, for "rna_denovo", instead of describing the parameter settings, the authors have simply dumped a line from the shell script into the manuscript – this is unacceptable.
14. Please explain the comment: "Any variant that had more than one mutation was included in the feature matrix twice" – does this mean that each variant is a different training example? If so, this approach seems to generate a bunch of issues with computing performance, stratified sampling for training / test split that are not discussed. Please clarify.

Reviewer #3 (Remarks to the Author):

An important challenge in understanding RNA molecular recognition is dealing with the interdependent nature of sequence and structure. For ADAR this problem is particularly complex since the enzyme recognizes multiple alternative substrates that differ in sequence and structure, and that are edited at different efficiencies. Liu et al. try to get a handle on this problem by generating large numbers of mutants in three different ADAR 1 editing sites using CRISPR technology, and then use a clever deep sequencing approach to quantify the extent of editing for each variants they are able to detect. The

correlations between sequence variation and editing level are largely expected based on previous work, but the authors are able to resolve finer detail due to the larger data set. The approach has the further advantage of measuring editing extent of different sequence variants in vivo and therefore reflects biological specificity. Despite sophisticated analytics the authors find that the data do not contain sufficient information to explain the variation of editing levels between the different substrates. Thus, in the end the authors fail in their attempt for develop a comprehensive, predictive model that integrates structure and sequence. Nonetheless, they strongly confirm general principles for ADAR 1 recognition, and provide important new refinements to what is already known. The experimental approach is elegant and the ability to generate such high density structure/function data sets is impressive and should be of significant interest to the broad community of RNA biologists.

Specific points

1. The manuscript describes a tour de force structure-function study and the results are impressive. However, the term "comprehensive" is accurate in describing the analysis of sequence and structure specificity. Comprehensive analyses would encompass enough sequence variants to unambiguously identify all specificity determinants. Clearly the ability to develop only substrate-specific models of specificity illustrates the current analysis falls short. The authors simply need to use more accurate language, it does not diminish what has been accomplished.
2. The authors models only explain ca. 70% of the variation in editing levels for the sequence variants in the test data sets. They authors should more clearly describe the potential factors that are likely to account for the failure of the model. Can any trends or features within the population of poorly predicted sequence variants be recognized?
3. Throughout the manuscript the authors assume that sequence variation proximal to the editing site does not result in global or extensive misfolding of the substrate. How would local sequence context and its potential to form alternative secondary structure affect the modeling results? Experimentally confirming the structures of single mutant variants that have large unexpected effects, or variants that are otherwise outliers would probably help refine the analysis.
4. Previous studies used transcriptomic analyses or high throughput sequencing combined with selex. How to the present results compare to what was learned from these previous studies. A more systematic and organized effort to put the new results in the context of previous work clarify what is new and would improve the impact of the manuscript for a general audience.
5. Can the data from previous studies mentioned, above, be used to further develop or validate the machine learning model? With the basic features of the modeling in hand, what factors prevents application to these larger data sets? Likely one aspect is the need to key in specific detail regarding secondary structure.
6. To what extent will the level of RNA expression affects the processing efficiently? Potential substrates compete for association with the ADAR enzyme and relative rates will depend on both their k_{cat}/K_m and their concentration. Low expression could lead to low editing, even though the k_{cat}/K_m is favorable relative to other substrates. Can some of the data, such as difference between the results for the three substrate RNAs, be explained by effects on levels of expression.
7. For the NIEL 1 and TTYH2 substrates multiple sequence variants are identified that have higher levels of editing compared to the WT sequence. What does analysis of this sub-population reveal regarding specificity determinants? What features makes them better?
8. Page 6 - The terms "Free Energy" and "All Stem Length" are used without definition, and the logic driving the analysis beginning on page 6 is unclear. The topic sentence states, "We reasoned that ... RA thermodynamics could also affect editing efficiency". But what aspect of RNA thermodynamics (folding?) is left for the reader at this point. Please clarify.
9. On page seven the authors state, "Structure features were derived at the editing site and adjacent regions..." what does derive mean in here specifically, please elaborate.
10. What assumptions are built into the machine learning model that could affect the outcome? It would help the reader for the authors to clearly spell out the assumptions and limitations of their modeling approach when it is introduced in the text.
11. The machine learning and analytics is interesting, but its is not clear whether much new is learned in the end. Despite the highly quantitative analysis, the conclusions remain general since the parameters of the model only account for a portion of the sequence variation effects on editing

observed in the data set. Indeed, the authors might consider just calculating a sequence probability logo based on the sequence variants of the most efficiently edited substrates. Most of the results are fairly obvious like (page 10) " The structure-similarity score of variants compared to WT feature (sic) was positively correlated with editing levels". The fact that there are variants with higher editing levels compared to WT is more interesting, especially from a design perspective (see above).

12. The last sentence on page 10, continuing on page 11 is confusing. The more extensive, although not comprehensive, nature of the present study does not seem likely to account for differences with previous studies. More likely, its because the current study is conducted in vitro and provides a unique window on biological specificity, which is influenced by many factors including the context of the concentration of the RNA substrate, the surrounding RNA sequence and structure at individual sites, and the influence of RNA binding proteins.

13. Regarding the probability score for formation of native structure- how would this be affected if there were alternative structures that were active in addition to the native structure, different classes or subsets of structure classes would affect the results in what way?

Response to Reviewers' Comments:

Reviewers' comments:

Reviewer #1 (Remarks to the Author):

Here the authors describe a screening approach for studying the effect of sequence changes in ADAR substrates with the goal of understanding (and predicting) cis-regulatory elements present in the RNA. With a clever CRISPR-based mutagenesis approach they introduce a large number of nucleotide changes at the genomic loci for three different ADAR substrates and use NGS to evaluate the effects these changes have on editing efficiency. They go on to employ machine learning to define the relationship between the various RNA sequence and structural features altered and the editing efficiency observed. They find that their models are reasonably effective at predicting editing levels within a group of closely related sequences, but are not effective at predicting editing levels for substrates that differ in sequence substantially from the training set. ADARs have received a lot of attention recently given their role in human disease and their use in directed editing applications. Understanding how substrate structure controls editing efficiency is a long standing problem in ADAR research. While certain features of an RNA substrate (e.g. substantial duplex structure, A-C mismatch, 5' and 3' nearest neighbor preference, etc) have been known for some time to be important for editing, it is not currently possible to accurately predict editing efficiency from RNA sequence and structure alone. Therefore, the topic of this paper is very important. I do question the impact of the study in its current form, however.

We very much appreciate the reviewer acknowledging the importance and difficulty of the topic we are addressing. We responded to the comments below.

Major comments:

1) The study involves the introduction of mutations in the genomic loci for three ADAR substrates and editing takes place with endogenous expression levels of these substrates and the ADARs (likely ADAR1 in HEK293T cells). All changes in editing levels are interpreted to arise from changes in the ADAR-RNA interaction. However, minimal consideration is given to how the introduced mutations may have altered other types of RNA processing events that will occur in these cells. For instance, do the sequence changes made to the NEIL1 substrate have any effect on splicing? Changes in splicing could have an indirect effect on editing efficiency. Or...do the sequence changes made in the AJUBA substrate have any effect on the transcript half life given its location in the 3' UTR? Sequence changes could also alter the interaction with competing RNA-binding proteins. Is it possible that sequence-dependent effects on the cellular metabolism of these RNAs are complicating the analysis and limiting the broad application of the predictive models? These issues should be addressed (or at least discussed).

We thank the reviewer for these very important points. We agree with the concerns that if sequence variation changes the RNA processing, the resulting difference in RNA abundance may screw the editing levels beyond the intrinsic RNA specificity. However, our newly included analysis (**Supplementary Fig. 2**) strongly argues against a significant effects of sequence variation in our libraries on RNA abundance, as we observe a strong correlation between the RNA coverage and gDNA coverage in our sequencing data. I.e., while the abundance of individual sequence variants varies, this can be explained by the CRISPR knock-in efficiency (reflected in

gDNA coverage) (**Supplementary Fig. 2a, 2d**). In addition, we observed no correlation between RNA abundance and editing levels (**Supplement Fig. 2b, 2e, 2g**), suggesting that the RNA abundance might not be a concern for the editing level we measured. We added these results in the text as:

*“The coverage of RT-PCR product for each variant was generally well correlated with the corresponding coverage of the product amplified from gDNA ($R^2 = 0.87$ for NEIL1 and $R^2 = 0.25$ for TTYH2) (**Supplementary Fig. 2a, 2d**), suggesting that the RNA abundance is generally not affected by the introduced variants. There is no correlation between the RNA or gDNA coverage with the editing level for all three substrates, which is consistent with previous reports^{24,25} that argues against potential influence of the substrate expression level on the editing level (**Supplementary Fig. 2b-2c, 2e-2g**).”*

In our experiment we had stringent conditions that the RNAs we measured are pre-mRNA populations from the nucleus extract. Therefore, we are measuring pre-splicing populations. We also agree that it is beyond the scope of the current study to dissect the important *trans* regulation by RNA binding proteins. We discussed the importance of future inquiries into unknown RNA processing effects as follows:

*“Because we observed no correlation of RNA abundance on editing level (**Supplementary Fig. 2**) and our experiment measures pre-mRNAs, our editing analysis is likely not affected by potential effects of sequence variation on RNA processing. Nevertheless, understanding the interplay between RNA editing and various RNA biology such as RNA processing pathways and RNA binding proteins presents an important question for future investigations.”*

2) The authors state in the abstract that this work “provides guidance for designing and screening of antisense RNA sequences that form dsRNA...for ADAR-mediated transcriptome engineering”. However, it is not clear from the discussion how this is true other than this study suggests that predicting important features doesn't really work very well for new sites so one should screen for effective guide strands instead of trying to use prediction. Can the authors use the tools they have generated to predict effective guide strand designs for a therapeutically relevant directed editing target site? If so, they should provide an example. If not, then the “provides guidance for designing” statement should be removed from the abstract.

We agreed and deleted this statement and modified the last sentence in the abstract to:

“Our integrative approach can be applied to larger scale experiment towards deciphering the RNA editing code.”

3) The authors state in the discussion that their results “provide new possible designs to mimic the highly selective and efficient editing observed in some natural ADAR substrates”. However, the examples they give for “new” designs are features that had already been shown to be important for ADAR reactivity. The authors should clearly describe any truly new features they have identified as important and general (not just useful for controlling editing on one substrate).

We agree that the points we intended to convey were not clear and we have revised the text as follows:

“Currently, the RNA engineering methods mainly use antisense guide RNAs (gRNA) that perfectly duplex with the target region except for an A:C mismatch at the editing site⁵⁰⁻⁵⁶. Our results strongly support additional imperfectly base-paired designs to mimic the highly selective and efficient editing observed in the natural ADAR substrates with complex structure features. Such features include.....”.

To clarify the new findings, we highlighted the new top general features we found in red in **Fig. 8a** and described those features in results and discussions. Perhaps more importantly than the individual features identified in our analysis, our data paint a picture that the *cis*-regulation landscape of ADAR editing is too complex to be explained by a short list of general features. Therefore, we revised the text to clarify these points in the discussion. The text is now changed to:

“Our key results can be summarized in four main points. First, we found alternative structures that can be equally or better edited than the wild-type structure (Fig. 6, Supplementary Fig. 7-9). Second, our models confirmed all known features identified by previous biochemical and transcriptomic studies (A:C mismatch at the editing site, the 5' and 3' nearest neighbors, the length and stability of the substrate including 5' stem length and 3' loop structure)^{14-17,22} and revealed previously unexplored features, including ensemble diversity and closing pairs (Figs. 7-8, Supplementary Table 8). Third, our substrate-specific machine learning models integrated diverse sequence and structural features to and quantitatively predict editing levels of new variants for a given target (Fig. 7d). Fourth, both general features and substrate specific features synergistically contribute to editing levels and the degree of contribution of each feature varies across different RNAs, suggesting complex and context-dependent cis-regulation of the editing landscape (Fig. 7e-7g).”

Minor comments:

- 1) Top of page 5: “pyridine” should be “pyrimidine”
- 2) Page 8, Figure 5d should be Figure 5e?
- 3) Figure 5e should be Figure 5f?
- 4) Fig. 5f should be Fig. 5g?

We thank the reviewer for pointing out these errors. The errors are corrected, and the figures are updated.

Reviewer #2 (Remarks to the Author):

Using a CRISPR/Cas9 strategy, this manuscript presents new data on the effect on editing rates of mutations to three human ADAR1 editing sites with the overall goal of better understanding ADAR1 targeting. Specifically, they measure the impact of some sequence and secondary structural features with CRISPR/Cas9-mediated mutagenesis and use a machine learning model to predict substrate-specific RNA editing levels and further investigate contributions of a set of 122 features. This model, PREUSS, when trained on data on some of these mutations, is able to predict with high accuracy, the impact of other mutations.

Better understanding of ADAR targeting would permit better design of antisense oligos to target specific RNAs for editing via endogenous ADAR pathways, potentially opening a new strategy for designing targeted gene therapies. Also, ADAR is a double-stranded RNA-binding protein, more insight into its target preferences

would also contribute to an overall understanding of how this important class of proteins identifies their target sites.

There has been some effort detailing sequence and structural preferences of ADAR targeting, this is the first attempt that I know of that comprehensively queries the impact of a large number of targeting mutations on editing rates at multiple targets. These data have the potential to inform detailed models of ADAR targeting that generalize beyond the three sites studied here.

Unfortunately, this manuscript does not achieve this goal. Specifically, the manuscript demonstrates, convincingly, that while it is possible to predict the impact of mutations to a target site that the classifier is trained on, these predictors do not generalize to other target sites. As such, it is not clear to me that the manuscript achieves any of the claims in the abstract and introduction. Specifically, it is not clear what additional new insight it provides about the general ADAR editing code (above what was already known and referenced in the manuscript) – the general editing code features shared across the three sites were already known, or seem vaguely defined. Nor it is clear what new guidance it provides for designing antisense RNA sequences for transcriptome engineering: if PREUSS is editing site specific, how will it help design antisense oligos for target sites it was not trained on? Finally, there seems to be little to recommend this modelling strategy as a foundation for developing predictive models of RNA editing – the design of the PREUSS features depend heavily on the WT site, what is the WT site for new sites for transcriptome engineering? What is the WT site when being used detect new editing sites generated by SNPs or somatic mutations? Also, there is an additional question of novelty of the modelling strategy compared to recent work on predicting efficacy of CRISPR guides (PMID: 26780180, PMID: 29998038, PMID: 28263296), this work is not cited in the manuscript and seems to be the most similar work, rather than the splicing code work that is cited.

In addition to these concerns, it appears that the manuscript includes an incorrect, or old, draft of the computational methods section. There are a number of editing errors and incorrect citations in that section, some of which I have detailed below. This section requires extensive revision before it is suitable for review.

We thank the reviewer for recognizing the comprehensive scales of our work and for the constructive comments. We agree that the ability of our model to predict editing for variants of a given target, but not across targets, is an important limitation. However, we also believe that this result from our systematic investigation—that the sets of features influencing editing vary greatly across targets, rather than being limited to a simple set of universal features, in an important one. We have edited the manuscript, including the abstract, to clarify the scope of the impact of our approach. We now also describe in greater detail the new properties identified as important for editing by PREUSS, both general and substrate specific.

The reviewer raises an excellent point that in many cases, the wild-type sequence of an RNA isoform may not be known, and the concept of a “mutation-associated feature” loses its biological meaning. **Computationally, however, the method itself does not require knowing which sequence is the WT. Any one of the RNA in a library can be called a “WT”.** Taking transcriptome engineering as an example, for a given target site, one can start from a perfect complementary antisense oligo and call the duplex formed between the target site and the oligo a “WT”. Then the variant information such as sequence variation can be encoded relative to the so-called “WT” for our supervised learning method. In other words, this so-called “WT” related feature is a way of choosing a reference point for generating computerized descriptions of features for supervised machine learning, which does not require the biological meaning of WT.

Nonetheless, we agree that certain features, such as the “number of mutations”, is a limitation of the current modeling approach. **With the availability of a larger dataset, it would be possible to train a convolutional neural network that avoids an *a priori* featurization of the data. We have followed this approach with the current dataset to demonstrate feasibility detailed below (Figure R1).** This proof-of-concept analysis suggests that learning ADAR editing levels directly from sequence and structure information, without using the number of mutations and associated metrics as features, is possible. The reason this approach was not used as the primary modeling tactic in the paper is its tendency to overfit to the training set when the dataset is small. However, this approach would be the method to use if a larger dataset of RNA isoforms was available. The method and results of this method is described below in detail.

Result of convolutional neural network

We performed an exploratory analysis to determine the feasibility of predicting ADAR editing levels from substrate sequence and structure directly without an *a priori* featurization (Figure R1a). The model was trained on two inputs, one-hot-encoded sequence and one-hot-encoded bpRNA structure, for each isoform. The model used a mean-squared-error (MSE) loss to learn the editing level from these two inputs. Hyperparameter search yielded an optimal architecture of one convolution layer with six filters, kernel size six, for each of the two inputs. The inputs were then concatenated and passed through a dense layer of output dimension size 8 (Figure R1a and b). The model achieved a training set performance of Spearman R = 0.82, Pearson R = 0.92, and a test set performance of Spearman R = 0.81, Pearson R = 0.83 (Figure R1c). SHAP importance scores were calculated for each input base and bpRNA structural annotation per test set isoform (representative examples of SHAP annotation for NEIL1, TTYH2, and AJUBA are indicated below (Figure R1d-f). High SHAP values were assigned to the base and structural elements at the mutation and editing sites. Other features with high SHAP scores matched those assigned high SHAP scores in the XGBoost analysis --i.e. the presence of a hairpin loop upstream of the editing site (Figure R1f).

Convolutional neural network training

The model was trained on a joint input of 924 isoforms across the three substrates, split at random into a training set (n=647), a validation set (n=139), and a test set (n=138). Isoform sequence and bpRNA structure were one-hot-encoded into a matrix representation. Given that the isoforms were of a non-uniform length, the one-hot-encoded structure and sequence matrices were padded with zeros to the length of the longest isoform in the dataset (166 bases) to achieve uniform dimensionality for all input elements.

The keras (v. 2.3.0) framework¹ with the tensorflow (v.2.1.0) backend² was used to perform hyperparameter search for the optimal convolutional neural network architecture. The hyperparameter combinations tested, and the resulting performance on the training and test sets, are provided in the table below. The optimal architecture consisted of a single one-dimensional convolution layer for each of the two inputs (structure and sequence). This layer contained 10 filters of kernel size 6. The outputs of the convolution layers were concatenated along the channel axis (encoding base and structural element) and passed through a single fully connected layer of output dimension 8. The model was trained with a mean squared error loss (MSE) using the Adam optimizer with a learning rate of 0.001, using a stopping criteria of no reduction in validation loss for three consecutive epochs. Each epoch consisted of a full pass through the training dataset.

The SHAP DeepExplainer tool³ was used to assign importance scores to each test set sequence and structure input. The full training set was used as the background for DeepExplainer. The DeepSHAP scores were projected on the sequence and structure input matrices and visualized with the Python logomaker library⁴.

Figure R1: Convolutional neural network predicts ADAR editing level directly from RNA isoform sequence and structure. a) Model architecture consists of one-hot-encoded sequence and bpRNA structure, a single convolutional layer with 10 filters, and a single fully-connected layer. b) Training and validation loss curves. c) Observed vs predicted ADAR editing levels on the joint NEIL1, TTYH2, AJUBA training and test sets. Points in the scatter plot are color-coded by how frequently the values were observed. The model achieved training set performance of Spearman $R = 0.82$, Pearson $R = 0.92$ and test set performance of Spearman $R = 0.81$, Pearson $R = 0.83$. d) SHAP importance scores for each sequence base and structural annotation in an example NEIL1 isoform. The blue triangle indicates the position of the ADAR

editing site; the red triangle indicates the position of the sequence mutation. e) SHAP importance scores for an example TTYH2 isoform. f) SHAP importance scores for an example AJUBA isoform.

Convolutional neural network hyperparameter tuning strategy.

The parameter values tuned were “filters” (number of convolutional filters in the convolutional layers of the model); “filter size” (kernel size of the filters in the convolutional layers of the model); “BatchNorm” (whether or not batch normalization was performed immediately following the convolutional layers); “MaxPool” (where or not a MaxPool layer of size 3 was applied after the convolutional layers); “BiLSTM” (whether or not a bi-directional LSTM unit was applied after the convolutional layers in the model); “Dense layer” (size of the output dimension of the dense layer in the model); . The table also includes the training and test set performance metrics (Spearman R, Pearson R), for each hyperparameter combination.

Filters	1	1	1	1	1	10	10	10	10	10	10	10	10
Filter size	6	6	6	6	12	6	6	12	6	6	6	6	6
BatchNorm	No	Yes	Yes	Yes	Yes	Yes	Yes	Yes	Yes	Yes	Yes	Yes	Yes
MaxPool	No	No	No	3	No	No	No	No	No	3	No	No	No
BiLSTM	No	No	No	No	No	No	No	No	No	No	128	64	256
Dense layer	32	32	64	32	32	32	16	16	8	8	No	No	No
Train Spearman	0.81	0.83	0.86	0.68	0.82	0.77	0.8	0.82	0.82	0.76	0.57	0.56	0.54
Train Pearson	0.86	0.87	0.91	0.76	0.86	0.85	0.89	0.91	0.92	0.87	0.6	0.62	0.57
Test Spearman	0.72	0.74	0.71	0.61	0.65	0.76	0.8	0.8	0.81	0.78	0.6	0.54	0.6
Test Pearson	0.77	0.78	0.76	0.72	0.67	0.81	0.83	0.83	0.83	0.82	0.63	0.58	0.62

Reference for convolutional neural network method

1. Chollet, F. & Others. keras. (2015).
2. Abadi, M. *et al.* TensorFlow: Large-Scale Machine Learning on Heterogeneous Distributed Systems. *arXiv [cs.DC]* (2016).
3. Shrikumar, A., Greenside, P. & Kundaje, A. Learning Important Features Through Propagating Activation Differences. in *Proceedings of the 34th International Conference on Machine Learning - Volume 70* 3145–3153 (JMLR.org, 2017).
4. Tareen, A. & Kinney, J. B. Logomaker: Beautiful sequence logos in python. doi:10.1101/635029.

To summarize, the featurization methods using WT as a reference point can still be applied to any editing site including for transcriptome engineering as discussed above. In addition, a convolutional neural network is promising with larger data set in future.

We now included suggested references and revised the discussion text.

We also corrected the editing and citation errors for the computational methods sections.

Detailed comments:

1. The authors mentioned that the CRISPR oligos used in the study include all possible single mutants and 11% of all possible double mutants for NEIL1. How were these double mutants selected? The authors provided a

heatmap of editing levels of double mutations in the editing strand of NEIL1 on figure 2d, and concluded that, for double mutants, at least one mutation close to editing site would result in stronger effects. Given that only 11% double mutants were tested and they are all manually selected, are those samples representative enough to generalize about the feature effects for double mutants?

Most of the double mutants are double-transversion mutants. This was deliberately designed to theoretically disrupt the secondary structure as much as possible so we can obtain a wide range of secondary structures within our library as to learn about structural features. We modified the text and figures to specify the double-transversion mutants.

We now clarify that the text “at least one mutation close to editing site resulted in stronger effects” refers to this specific dataset, rather than a general conclusion.

2. For secondary structure prediction and mutation type annotation of AJUBA, the manuscript uses a dsRNA stem formed by a complementary sequence 500+nt away from the editing sequence. Is this the verified ECS for the editing site? How did the author choose this ECS?

We defined the ECS site as reported previously (see citations in the Methods), by examining the 100 to 1000 bp sequence flanking the editing site and choosing the most stable RNA duplex. This predicted ECS also matches the prediction from the RNAhybrid method suggested by in point 3 below (**Figure R2**).

3. In particular, the AJUBA site seems particularly problematic. Perhaps some of the unique properties of mutations to the editing site are because there are other potential ECSs within the 500nt gap besides the proposed one, so mutations to the editing site will influence the choice of ECSs. The folding strategy for the AJUBA site is problematic too because it requires inventing a loop sequence which does not exist. Perhaps a tool like RNAhybrid would be more appropriate?

Thanks for pointing out the useful tool RNAhybrid that for searching RNA duplex targets in a long sequence (such as for miRNAs). The AJUBA RNA omitting the rest of the long-range sequences (**Fig. 1b**) was chosen up to a region where a loop structure naturally exists that mimics a hairpin structure when folding the whole gene. Using the suggested RNAhybrid web server, we also used the editing strand shown in **Fig. 1b** to search for a “target site” in the AJUBA gene. The resulting prediction completely matches our prediction as shown below (**Figure R2**). We now mention this validation with the RNAhybrid prediction in the Methods:

“We defined the ECS site as reported previously, by examining the 100 to 1000 bp sequence flanking the editing site and choosing the most stable RNA duplex. This AJUBA ECS sequence also matches the predicted duplex region using RNAhybrid.”

```

Version: RNAhybrid 2.2
Command line: /vol/bioapps/bin/RNAhybrid.bin -q
/var/bibiserv2/anonymous/rnahybrid/25/24/40/bibiserv2_2020-06-
25_004037_KQZKe/rnahybrid_input_mirna_sequences.file -m 710 -t
/var/bibiserv2/anonymous/rnahybrid/25/24/40/bibiserv2_2020-06-
25_004037_KQZKe/rnahybrid_input_target_rna_sequences_.file -n 50 -D -b 1
searching
dataset: 1
mde of editing_strand: -112.000000
Individual hits
-----

dataset: 1
target: AJUBA
length: 710
miRNA : editing_strand
length: 50

mfe: -77.4 kcal/mol
p-value: undefined

position 127
target 5' A           U           G           G C           A           G 3'
          GUUUUGGGUUGUGGU GAUGCAGU UGGGAU UC CUGAGAGGU GCAA
          CAAAACCCUAACACUA CUACGUUA GCCCUG GG GACUCUCCA CGUU
miRNA 3'           C           G           G A           C           5'
-----

```

Figure R2. RNAhybrid prediction for ECS sequence of AJUBA. This prediction validate the ECS sequence we computed in the work.

4. ECS is an awkward abbreviation especially because “editing site” is not abbreviated

ECS is historically widely used in the RNA editing field and the “editing strand” is not abbreviated to avoid confusion between “editing strand” and “editing site”.

5. Why is it important to choose target sites in different transcript regions? This was not clear to me in the motivation.

We clarified this point in the discussion as “*This showcases that a screen of possible designs of gRNA would be a valuable and cost-effective strategy to learn the best features that lead to the most specific and efficient editing for each different target site in transcriptome engineering.*” We discussed this in the context of emerging transcriptome engineering efforts analogous to gene therapy.

6. It seems strange that three mutations abolish editing in all cases, but some cases of 4 and 5 mutations do not. Why do you think this is?

In case of TTYH2, several sets of 4–5 mutations indeed preserved editing (**Supplementary Fig. 1b**). Inspection of these variants reveals that these mutations maintain well duplexed 5’ stem region or mimic the loops observed in the wild-type TTYH2 structure.

7. What do you mean by “using dsDNA as a donor” (main text, near reference to Supp Fig 1j-k?)

To efficiently introduce point mutations by CRISPR/Cas9 knock-in, a donor oligo is needed together with a sgRNA to utilize the homology directed repair (HDR) pathway. We revised the text to clarify the advantages of using dsDNA vs. ssDNA as donor oligo:

“Interestingly, we discovered that similar knock-in efficiency and editing results were achieved when using ssDNA oligonucleotides or dsDNA (e.g., PCR product) as the donor for CRISPR-mediated HDR (Supplementary Fig. 1g-1j). Using dsDNA PCR products greatly simplified the procedures and reduced the cost of the experiments.”

8. The notation “transversion” and “transversion+break” is confusing. Why wouldn’t a transversion always cause a break? It seems like what is meant by transversion is that the base altered is in a loop. If so, the notation could be clearer here.

We thank the reviewer for pointing out the confusing terminology. An example of transversion that doesn’t cause base-pair break is now shown in the new **Fig. 3a**, where the mutation was located in a bulge that didn’t change the structure.

Additionally, after re-examining the data, we realized that we previously didn’t separately address the scenario where a mutation leads to base-pairing with a new partner nucleotide. Therefore, in this revision we added two new categories (‘transition + shift’ and ‘transversion + shift’) to account for structural changes other than a simple base-pair break at the mutation site. Examples of all 6 categories are now provided in a new figure (**new Fig. 3a**). The according texts regarding these changes are revised but the main conclusions remain the same.

9. The claim that “primary sequence might have a larger influence on editing of the AJUBA RNA” seems unsupported by the data. In particular, as I mentioned above, because it is such a long range interaction, there could be an alternative ECS which is favored by the transitions.

We agree that the long-range interaction could result in AJUBA being more sensitive to mutations. As described earlier point in 2 and 3, computationally we have reasonable confidence for the ECS we predicted for AJUBA. Therefore, we modified the text as:

“Many AJUBA single mutations have much larger effects (z -score < -3) than single mutations in NEIL1 and TTYH2. This difference may be explained by the long (524 nt) distance between AJUBA editing strand and ECS, such that mutations may lower the probability of forming this long-range structure relative to alternative proximal structures; alternatively, primary sequence might have a larger influence on editing of the AJUBA RNA.”

10. Also, it is not clear how to judge whether a difference between -71 kcal/mol and -87 kcal/mol is meaningful. Presumably, if both free energies are high enough to ensure stable dsRNA formation, there is no real functional difference between them.

We agree and have deleted this statement.

11. The claim: “these observations indicate that RNA primary sequence and structure are intertwined in terms of mediating effects of mutations...” seems vague. What would be the alternative be? Isn’t this what you would

have already expected given the prior work on ADAR targeting? Furthermore, in the next paragraph, there is a further claim that “mutations may need to impact secondary structure to have major effects on RNA editing efficiency” which seems to invalidate the previous claim that there is an interplay.

We agree that the original texts were misleading. We changed this sentence to “*these results are consistent with previous observations of intertwined sequence and structure effects on editing....*”.

In the next paragraph we deleted the sentence to clearly state the observations from the compensatory mutations without jumping to conclusions.

12. The fact that number of mutations from the WT site is the most predicted feature is particularly disappointing as it doesn't really provide insight into the editing code nor permit generalization to new sites. Why was this feature used? What happens if it is not included?

We included the ‘number of mutations’ as a measure of sequence similarity to the wild-type construct, analogous to the measures of structural similarity, although we agree that there is overlap between these features, which complicates physical interpretation. **Figure 8** and **Supplemental Figure 10d** show the relative contributions of all feature sets to the predicted editing levels, and suggest that the model still achieves reasonable performance when mutation-related features are omitted. If the sole omitted feature is the “number of mutations” in the substrate, the remaining features account for 88% of the contribution to the model's predictions in the joint training method (see “no_num_mutations” in **Supplemental Figure 10d**).

13. The computational methods section needs editing. First of all, it is incomplete and details needed to reconstruct the methods are missing. Also, the citations are incorrect: e.g. the bpRNA is not described in citation 30. Finally, it is repetitive: we are told twice in the same paragraph that a temperature of 37C was used with RNAfold. The constraint description for the three target sites do not seem correct – why would one force the 5' and 3' stem flanking regions to be unpaired for NEIL1, TTYH2? Further, for “rna_denovo”, instead of describing the parameter settings, the authors have simply dumped a line from the shell script into the manuscript – this is unacceptable.

Thank you for the feedback. We have modified the methods section to provide additional details. We now also provide Jupyter notebooks in our github repository that fully reproduce the XGBoost modeling analysis, and have provided increased documentation in the github repo detailing the feature generation, bpRNA structure determination, and other computational analyses (<https://github.com/kundajelab/PREUSS>).

The bpRNA citation has been corrected in the main text.

The duplicate text about 37°C in that paragraph has been removed.

We clarified the constraints description in the Methods. The dots in the dot-bracket notation for the secondary structure constraints indicate that no constraint is applied at those positions. Specifically, we explain that no additional penalties were added for base pairs that may form outside the core region of the RNA as follows: "*Note that penalties are not applied for any additional base pairs that form in the unpaired regions of the reference secondary structures below.*"

We have additionally clarified the methodology used for RNP-denovo modeling. All non-default options are now described before the command line script.

14. Please explain the comment: “Any variant that had more than one mutation was included in the feature matrix twice” – does this mean that each variant is a different training example? If so, this approach seems to generate a bunch of issues with computing performance, stratified sampling for training / test split that are not discussed. Please clarify.

Yes, each variant constitutes a different training example, as we hypothesized that different combinations of variants within an isoform sequence would produce different effects on editing level. Consequently, a number of the features used for training are based on the position and type of variant. These are documented in Supplementary Table 1 (the “mut_*” categories of features). We verified that there were no cases when an isoform could appear in multiple folds (i.e., no train/test contamination) due to having multiple mutations (all mutations for a given isoform would be exclusively in the training set or exclusively in the validation set or exclusively in the test set). This clarification has been added to the Methods section.

Reviewer #3 (Remarks to the Author):

An important challenge in understanding RNA molecular recognition is dealing with the interdependent nature of sequence and structure. For ADAR this problem is particularly complex since the enzyme recognizes multiple alternative substrates that differ in sequence and structure, and that are edited at different efficiencies. Liu et al. try to get a handle on this problem by generating large numbers of mutants in three different ADAR 1 editing sites using CRISPR technology, and then use a clever deep sequencing approach to quantify the extent of editing for each variants they are able to detect. The correlations between sequence variation and editing level are largely expected based on previous work, but the authors are able to resolve finer detail due to the larger data set. The approach has the further advantage of measuring editing extent of different sequence variants in vivo and therefore reflects biological specificity. Despite sophisticated analytics the authors find that the data do not contain sufficient information to explain the variation of editing levels between the different substrates. Thus, in the end the authors fail in their attempt for develop a comprehensive, predictive model that integrates structure and sequence. Nonetheless, they strongly confirm general principles for ADAR 1 recognition, and provide important new refinements to what is already known. The experimental approach is elegant and the ability to generate such high density structure/function data sets is impressive and should be of significant interest to the broad community of RNA biologists.

We appreciate the reviewer for the positive comments and for recognizing the significance of our approach to apply to broad RNA biology. We also agree that our current predicative models are limited to specific substrates. We revised the texts accordingly as detailed below for each point.

Specific points

1. The manuscript describes a tour de force structure-function study and the results are impressive. However, the term “comprehensive” is accurate in describing the analysis of sequence and structure specificity. Comprehensive analyses would encompass enough sequence variants to unambiguously identify all specificity determinants. Clearly the ability to develop only substrate-specific models of specificity illustrates the current

analysis falls short. The authors simply need to use more accurate language, it does not diminish what has been accomplished.

We are grateful for the reviewer's positive comments and very helpful suggestions. We no longer refer to the analysis as comprehensive and have substantially revised the language according to the reviewer's suggestions.

2. The authors models only explain ca. 70% of the variation in editing levels for the sequence variants in the test data sets. They authors should more clearly describe the potential factors that are likely to account for the failure of the model. Can any trends or features within the population of poorly predicted sequence variants be recognized?

To address the reviewer's comments about potential factors limiting our modeling, we discussed (1) ways to extend our work to overcome the experimental limitations in the paragraph starting with "*Our approach opens several new lines of inquiry for further improvement.*", and (2) pointed out limitations of the modeling method and future directions in the paragraph starting with "*There are several limitations to the modeling approaches utilized in this study.*"

The reviewer raises an interesting point about what can be learned from just looking at the poorly edited substrates. We plotted the feature value distributions for the isoforms with Z-score of editing level difference from wild type ≥ 1 , Z-score of editing level difference from wild type ≤ -2 , and Z-score of editing level difference from wild type in the range $(-2 < Z < 1)$. A fairly liberal threshold of $Z=1$ was used for highly edited isoforms as no isoforms were found to have a difference in editing level from wild type with Z-score > 2 . The resulting distribution, shown below (**Figure R3**), suggests no systematic individual feature trends unique to especially highly or lowly edited isoforms. In addition, as shown in the new clustering analysis (**Fig. 6, Supplement Fig 7 to 9**), there are many different combinations of structures and sequence for the poorly or highly edited variants. **A generalized list of one-feature-fit-all effects are not readily describable. This result reinforces our observation that no single feature fully explains variations in editing level; rather a combination of features jointly explains why some isoforms have high editing and others do not.** This is precisely why the machine learning approaches (XGBoost in the manuscript and convolutional neural network analysis in **Figure R1 above**) that use a combined set of multiple features and/or learn a featurization directly from sequence and structure allow us to more accurately predict editing levels than examination of individual features independently.

Figure R3: Features value distributions of different groups of editing substrates.

3. Throughout the manuscript the authors assume that sequence variation proximal to the editing site does not result in global or extensive misfolding of the substrate. How would local sequence context and its potential to form alternative secondary structure affect the modeling results? Experimentally confirming the structures of single mutant variants that have large unexpected effects, or variants that are otherwise outliers would probably help refine the analysis.

We agree with the reviewer that our previous analysis largely focused on using the structure features that derived from the MFE (minimum free energy) structure. Two features we previously included, free energy (ensemble free energy) and the probability of active conformation, can partially reflect global folding properties of the RNAs, In terms of alternative structure for the modeling, we now included three additional thermodynamic parameters, minimum free energy, ensemble diversity (calculated from the base-pairing probabilities from partition function folding which can reflect the extent of alternative structure formation), and MFE frequency (indicating the stability of the MFE structure in the ensemble) (**Fig. 5**). These features serve as further quantitative parameters reflecting global RNA folding properties and are also ranked high in the XGBoost machine learning models (**Fig. 7-8**).

In terms the sequence context, our model included sequence features that were defined using either WT, mutation, and editing site as reference, such as `mut_ref_nt`, `mut_nt`, `site_prev_nt` and `site_next_nt` (see **Supplement table 1**) and some of those features are ranked high in our XGBoost models (**Fig. 7g**).

In terms of using experimentally determined RNA structure, we were able to use DMS chemical probing for all NEIL1 variants and a portion of TTYH2 RNAs (TTYH-ECS) (added in the Methods). The computationally predicted structure largely agrees with structures inferred from DMS chemical probing as shown (similarity score close to 1 suggesting experimentally inferred RNA structure largely agrees with the computational prediction, **Supplement Fig. 4c**). Using these data did not improve our modeling results. Therefore, considering that not all labs applying our PREUSS pipeline can measure RNA structures experimentally, we stick to the computational predicted structures to keep consistency of data analysis across all substrates. Although the *in vitro* structure mapping is informative, the best experimental validation of these RNA structures would be conducting DMS-MaPseq in cells as we stated in the discussions for future directions.

4. Previous studies used transcriptomic analyses or high throughput sequencing combined with selex. How to the present results compare to what was learned from these previous studies. A more systematic and organized effort to put the new results in the context of previous work clarify what is new and would improve the impact of the manuscript for a general audience.

We appreciate these comments. To clearly present our results in the context of previous work and highlight what's new, we colored the new top general features we found in red in the **Fig. 8a** and revised the discussions as:

*“Our key results can be summarized in four main points. First, we found alternative structures that can be equally or better edited than the wild-type structure (**Fig. 6, Supplementary Fig. 7-9**). Second, our models confirmed all known features identified by previous biochemical and transcriptomic studies (A:C mismatch at the editing site, the 5' and 3' nearest neighbors, the length and stability of the substrate including 5' stem length and 3' loop structure) and revealed previously unexplored features, including ensemble diversity and closing pairs (**Figs. 7-8, Supplementary Table 8**). Third, our substrate-specific machine learning models*

integrated diverse sequence and structural features to and quantitatively predict editing levels of new variants for a given target (Fig. 7d). Fourth, both general features and substrate specific features synergistically contribute to editing levels and the degree of contribution of each feature varies across different RNAs, suggesting complex and context-dependent cis-regulation of the editing landscape (Fig. 7e-7g)."

In terms of the previous SELEX analysis, it used dsRBD of *Xenopus* ADAR1 and identified tight binders to dsRBD enriched with tripartite stem-loop structures in which the two short 4-6 bp stems are preferably bound by dsRBD and longer stem also interact with the protein. Compared to the dsRBD binders from SELEX, our highly edited RNA substrates are largely a single stem-loop shape, while the multi-stem-loop shaped RNAs are poorly edited (Fig. 6b). Our studies focused on the RNA editing activity which are largely determined by the deaminase domain but the dsRBD also contributes to substrate recognition. Therefore, an important future direction would be to conduct high-throughput *in vitro* RNA binding experiments for human ADAR1 together with the existing SELEX data to use our computational methods to dissect features that are more specific to dsRBD. We added these texts to discuss this point as follows:

*"Although our data focused on the editing level, which is largely determined by the deaminase domain^{17,44}, the dsRBD (illustrated in Fig. 5c) also contributes to substrate recognition⁴⁵⁻⁴⁷. In the future, high-throughput *in vitro* RNA binding experiments can be performed to combine with existing SELEX data for dsRBDs⁴⁸ to identify features specific to dsRBDs of human ADARs using our pipeline."*

5. Can the data from previous studies mentioned, above, be used to further develop or validate the machine learning model? With the basic features of the modeling in hand, what factors prevents application to these larger data sets? Likely one aspect is the need to key in specific detail regarding secondary structure.

Very insightful discussions. Our current supervised learning requires pre-selected features and the values of certain features can be substrate specific, thus is best suited for systematically engineered variants. Transcriptomic data has the benefit of large amounts of editing sites for various substrates. But there are several factors that prevent direct application of our current machine learning models. First, the transcriptome data is less suitable for systematic study like ours due to the limited number of natural variants and the diverse nature of different substrates in the genome. Interestingly, our attempts of the neural network method (Figure R1) suggests that with larger datasets, we might not be limited by the featurizations. Second, as pointed out by the reviewer an important factor that prevents direct application of our modeling is we are still limited by the accuracy of the RNA structure features for most endogenous substrates due to the fact that the majority of the ECS are not well defined in human cells nor easily defined computationally in human transcriptome. Recently, irCLASH method was developed to define *in vivo* ADAR substrates duplexes (Song et al, Nature Structural & Molecular Biology volume 27, pages351–362(2020)), which would be a good data source to apply our analysis in future. But certain representative editing substrates such as the AJBUA we tested was not presented in the irCLASH data, highlighting technical aspects that could be improved in the future to aid application of our PREUSS pipeline to transcriptomic data.

6. To what extent will the level of RNA expression affects the processing efficiently? Potential substrates compete for association with the ADAR enzyme and relative rates will depend on both their k_{cat}/K_m and their concentration. Low expression could lead to low editing, even though the k_{cat}/K_m is favorable relative to other

substrates. Can some of the data, such as difference between the results for the three substrate RNAs, be explained by effects on levels of expression.

It is established in the literature that there is no correlation between the level of RNA expression and the editing efficiency (PMID: 28266523 and PMID: 29592874). The three RNAs we choose all require gene specific primers to amplify the RNA during the RT step, so our data could not show the expression level of the genes. However, we show that the coverage of RNA in our data does not correlate to the editing levels (**Supplement Fig. 2**), arguing against a simple way that the relative RNA concentration affects the editing levels we measured. We cannot rule out other complex ways the RNA variants can affect RNA half-life and interactions with other biological pathways. In future, measuring multiple time points during the CRISPR experiments and using *in vitro* assay for measuring the kinetics of editing to extrapolate k_{cat}/K_m values would deepen our understanding of the substrate specificity. We added discussions about these limitations:

*“Because we observed no correlation of RNA abundance on editing level (**Supplementary Fig. 2**) and our experiment measures pre-mRNAs, our editing analysis is likely not affected by potential effects of sequence variation on RNA processing. Nevertheless, understanding the interplay between RNA editing and various RNA biology such as RNA processing pathways and RNA binding proteins presents an important question for future investigations.”*

7. For the NIEL 1 and TTYH2 substrates multiple sequence variants are identified that have higher levels of editing compared to the WT sequence. What does analysis of this sub-population reveal regarding specificity determinants? What features makes them better?

Please see response to point number reviewer #2's point 2 above -- we have plotted the feature distribution for isoforms with higher ($Z > 1$) editing levels relative to wild type. No single feature accounts for this difference (**Figure R2**), but a combination of effects across a number of features account for differences in editing level (**Figure 7 and 8**). The features that account for the highest percentage of difference in editing level are presented in **Figures 7g and 8a**.

8. Page 6 - The terms “Free Energy” and “All Stem Length” are used without definition, and the logic driving the analysis beginning on page 6 is unclear. The topic sentence states, “We reasoned that ... RNA thermodynamics could also affect editing efficiency”. But what aspect of RNA thermodynamics (folding?) is left for the reader at this point. Please clarify.

To clarify the terms, we revised the texts as below:

“We reasoned that structural and thermodynamic features affecting RNA stability could also affect editing efficiency.”

9. On page seven the authors state, Structure features were derived at the editing site and adjacent regions...” what does derive mean in here specifically, please elaborate.

We elaborated this in the **Methods** section (shown below) and in the Supplementary Table 1.

“Feature extraction

*RNA structures for NEIL1, AJUBA, and TTYH2 were annotated with the bpRNA algorithm⁹. The bpRNA annotations were in turn utilized to extract structural and positional features for each variant. A feature matrix with structure-specific features from the bpRNA annotations, sequence-specific features, features that take into account the isoform mutation type and position, and thermodynamic-specific features was engineered (how the features were derived are described in **Supplementary Table 1**) for each substrate and included a total of 122 features (**Supplementary Tables 2, 3, 4**). ”*

10. What assumptions are built into the machine learning model that could affect the outcome? It would help the reader for the authors to clearly spell out the assumptions and limitations of their modeling approach when it is introduced in the text.

Several assumptions were in play in data featurization and generation of training/test/validation splits for modeling. These included the following:

- 1) The train-test-validation split for XGBoost analysis ensured that isoforms with mutations at a given sequence position were all assigned to a single split, to avoid train/test contamination. We made the assumption that this approach would be sufficient to ensure contamination of the splits did not occur. This is stated in the methods section:
“The dataset was randomly separated into 3 splits: training on 70% of variants, model validation on 15%, and testing on the remaining 15%. To avoid train/test contamination, base pair positions along the RNA molecules were assigned to one of the 3 splits (training, tuning, or test). All features associated with a given base pair position were assigned to the corresponding split.”
- 2) In determining our data featurization for XGBoost, we decided to look at sequence and structural elements up to a certain distance from the editing and mutation sites (3 structural / sequence elements upstream and downstream). We assume that considering 3 upstream/downstream elements will capture most of the mechanisms driving editing level. We state in the text *“as these regions within the RNA substrate fully encompass the interaction site with the ADAR deaminase domain (**Fig. 7c**)^{29,30}”*.
- 3) To some level, we assume that the featurization we are using for XGBoost will capture a combination of substrate-specific as well as substrate-agnostic features. However, this is an assumption that we validate experimentally by training both jointly across the three substrates and also training on two of the substrates and testing on the third.

11. The machine learning and analytics is interesting, but it is not clear whether much new is learned in the end. Despite the highly quantitative analysis, the conclusions remain general since the parameters of the model only account for a portion of the sequence variation effects on editing observed in the data set. Indeed, the authors might consider just calculating a sequence probability logo based on the sequence variants of the most efficiently edited substrates. Most of the results are fairly obvious like (page 10) “The structure-similarity score of variants compared to WT feature (sic) was positively correlated with editing levels”. The fact that there are variants with higher editing levels compared to WT is more interesting, especially from a design perspective (see above).

We greatly appreciate the suggestions on making the “take-home” message clear in our manuscript. Please see the response to point 4 above.

A simple sequence logo analysis would be less applicable in our dataset. To do a sequence logo analysis, the best kind of data would be maintaining the base pairing of the RNA structure when making mutations to isolate the sequence preference effects (the “triplet motif”) as meticulously demonstrated before by the Bass lab (Eggington et al, 2011). Overall, the triplet motif preference is not very strong compared to many other nucleic acid acting enzymes. The triplet motif effect in our data is significant but also small compared to the predictive power from other features (**Fig. 8b**).

We strongly agree that the RNAs that are edited higher than WT are very attractive. Our newly added RNA clustering analysis helps visualize the structures of those RNAs (**Fig. 6, Supplement Fig. 7-9**). Our machine model also picked up features learned from those substrates as discussed in the text. However, as discussed in the response to point 2, none of the single features show significant trends. Because it is not individual features but the synergetic interplay of features that determine editing levels, our machine model is more useful in that it can quantify how much these features contribute to editing level (SHAP values) under different context.

12. The last sentence on page 10, continuing on page 11 is confusing. The more extensive, although not comprehensive, nature of the present study does not seem likely to account for differences with previous studies. More likely, its because the current study is conducted in vitro and provides a unique window on biological specificity, which is influenced by many factors including the context of the concentration of the RNA substrate, the surrounding RNA sequence and structure at individual sites, and the influence of RNA binding proteins.

We agreed and deleted the sentence because it is confusing.

13. Regarding the probability score for formation of native structure- how would this be affected if there were alternative structures that were active in addition to the native structure, different classes or subsets of structure classes would affect the results in what way?

We clarified the scope of the probability score in the Methods.

“These constraints and reference information ensure that the “active conformation” we are calculating includes a group of conformations that closely resembles the MFE structure of the WT structure but are not limited to the WT MFE structure.”

Additionally, alternative structures exist (**Figure 6 and Supplementary Figure 9**) and our machine learning models do take into account possible alternative structures through multiple structures features as we discussed above and in the revised texts.

REVIEWER COMMENTS

Reviewer #1 (Remarks to the Author):

The authors have satisfactorily addressed the concerns I raised in my original review.

Reviewer #3 (Remarks to the Author):

Overall the authors have done an excellent job of taking into consideration reviewer comments and revising the manuscript accordingly. They acknowledge the limitations noted by all three reviewers and have edited the text to more accurately reflect the strengths and weaknesses of their model of ADAR specificity.

1. The authors have revised the text to account for suggestions to improve accuracy and clarity, in this case regarding the description of the work as "comprehensive". They have made a extensive efforts overall to accommodate reviewers comments in this regard.
2. Several limitations of the model forwarded by the authors were noted by all three reviewers. Here, the issue is the fact that the model does not account for all of the effects of sequence variation of editing. In response to this comment the authors analyze the poorly performing sequences but find not specific trend to suggest why they are not accounted for by the model. I agree the complexity of the factors involved precludes a simple answer. Nonetheless, accounting for the limitations noted by myself and the other reviewers this work represents an important step along the way toward a comprehensive model of specificity.
3. It is possible that surrounding RNA sequence influence the observed sequence specificity by forming alternative RNA structures. The authors have done their best to account for this issue by adding additional computational analyses applied to all sequences in their model and have confirmed several sequences using DMS probing. There is unlikely that more can be done at this point in the current manuscript. Inhibition arising from formation of alternative pairing interactions in ADAR substrates will need to await detailed biochemical studies.
4. The authors have included additional discussion relating their results to previous studies of ADAR specificity by SELEX. Although these data cannot be included in the current model it remains a potential source of training information.
5. See above
6. This issue was raised by both other reviews, they have made significant efforts to address issues of expression or alternative effects on RNA processing.
7. Unfortunately, no single feature nor additional information can be extracted from analysis of the sequences with higher scores than the native sequence.
8. The authors have edited the text to provide clarification
9. The authors have edited the methods section to clarify the meaning of "derive structural features"
10. The assumptions that are inherent in the machine learning model could still be better articulated. However, the authors have made a reasonable and acceptable effort to respond.
11. The suggestion was to see whether an optimal sequence logo might reveal features of optimal substrates, but it must be acknowledged the complexity of the model precludes a simple answer of this nature.
12. The text has been edited to improve clarity.
13. The issue here is essentially the same as #3, above. The ability of complex RNA motifs to form alternative structures by forming interactions with the upstream of downstream sequences is well known. However, there is no simple way to account for such effects other than to be diligent about considering whether stable alternative structures can be predicted, which the authors have done.

Reviewer #4 (Remarks to the Author):

As a fourth reviewer, I am just evaluating if the authors properly replied to the concerns raised by my colleagues.

The first concern from Reviewer 1, which I find very important, has not been properly addressed. By looking at general RNA levels, the authors are not looking into specific effects on splicing per se. Moreover, this statement is wrong: "In our experiment we had stringent conditions that the RNAs we measured are pre-mRNA populations from the nucleus extract. Therefore, we are measuring pre-splicing populations". Splicing occurs co-transcriptionally in the nucleus...the RNAs present in the nuclear fraction (RIBO 0 RNA-seq included) are already spliced unless the authors are evaluating specifically nascent RNA (kind of PRO-seq), which I don't think so. Splicing and RNA half-lives have to be properly addressed.

Regarding the response to Reviewer 2, I am not a neuronal network expert, I cannot thus evaluate if what the authors suggest is correct. But I can say that the authors failed to mention in what sense their manuscript has an added value respect the research papers cited: PMID: 26780180, PMID: 29998038, PMID: 28263296. I don't think point 14 is properly addressed either and it is an important point too.

Unless my colleagues say otherwise, I recommend that these points are further addressed before acceptance.

REVIEWER COMMENTS

Reviewer #1 (Remarks to the Author):

The authors have satisfactorily addressed the concerns I raised in my original review.

We appreciate reviewer 1 for considering our response and analyses satisfactory, particularly to address the concerns about splicing and RNA half-life.

Reviewer #3 (Remarks to the Author):

Overall the authors have done an excellent job of taking into consideration reviewer comments and revising the manuscript accordingly. They acknowledge the limitations noted by all three reviewers and have edited the text to more accurately reflect the strengths and weaknesses of their model of ADAR specificity.

1. The authors have revised the text to account for suggestions to improve accuracy and clarity, in this case regarding the description of the work as “comprehensive”. They have made a extensive efforts overall to accommodate reviewers comments in this regard.
2. Several limitations of the model forwarded by the authors were noted by all three reviewers. Here, the issue is the fact that the model does not account for all of the effects of sequence variation of editing. In response to this comment the authors analyze the poorly performing sequences but find not specific trend to suggest why they are not accounted for by the model. I agree the complexity of the factors involved precludes a simple answer. Nonetheless, accounting for the limitations noted by myself and the other reviewers this work represents and important step along the way toward a comprehensive model of specificity.
3. It is possible that surrounding RNA sequence influence the observed sequence specificity by forming alternative RNA structures. The authors have done their best to account for this issue by adding additional computational analyses applied to all sequences in their model and have confirmed several sequences using DMS probing. There is unlikely that more can be done at this point in the current manuscript. Inhibition arising from formation of alternative paring interactions in ADAR substrates will need to await detailed biochemical studies.
4. The authors have included additional discussion relating their results to previous studies of ADAR specificity by SELEX. Although these data cannot be included in the current model it remains a potential source of training information.
5. See above
6. This issue was raised by both other reviews, the have made significant efforts to address issues of expression or alternative effects on RNA processing.
7. Unfortunately, no single feature nor additional information can be extracted from analysis of the sequences with higher scores than the native sequence.
8. The authors have edited the text to provide clarification
9. The authors have edited the methods section to clarify the meaning of “derive structural features”
10. The assumptions that are inherent in the machine learning model could still be better articulated. However, the authors have made a reasonable and acceptable effort to respond.
11. The suggestion was to see whether an optimal sequence logo might reveal features of optimal substrates, but it must be acknowledged the complexity of the model precludes a simple answer of this nature.
12. The text has been edited to improve clarity.
13. The issue here is essentially the same as #3, above. The ability of complex RNA motifs to form

alternative structures by forming interactions with the upstream of downstream sequences is well known. However, there is no simple way to account for such effects other than to be diligent about considering whether stable alternative structures can be predicted, which the authors have done.

We appreciate reviewer 3 for acknowledging our responses in great details. As pointed out by the reviewer in each point above, our revisions and analyses in previous response addressed the reviewer's concerns satisfactorily and clarified many aspects requested by the reviewer.

Reviewer #4 (Remarks to the Author):

As a fourth reviewer, I am just evaluating if the authors properly replied to the concerns raised by my colleagues.

The first concerned from Reviewer 1, which I find very important, has not been properly addressed. By looking at general RNA levels, the authors are not looking into specific effects on splicing per se. Moreover, this statement is wrong: "In our experiment we had stringent conditions that the RNAs we measured are pre-mRNA populations from the nucleus extract. Therefore, we are measuring pre-splicing populations". Splicing occurs co-transcriptionally in the nucleus...the RNAs present in the nuclear fraction (RIBO 0 RNA-seq included) are already spliced unless the authors are evaluating specifically nascent RNA (kind of PRO-seq), which I don't think so. Splicing and RNA half-lives have to be properly addressed.

Regarding the splicing and RNA half-life questions, Reviewers 1 and 3 who previously raised those questions found our previous response satisfactory (see reviewer 1's response and reviewer 3's comments on point 6 above). We further clarify these points below.

We appreciate reviewer 4's comments and further clarify the details about our experimental conditions. Besides using the nuclear extract our experiment is also carried out so that only pre-mRNA was amplified for editing analysis. We now made this clear in the methods as: "*The primers were designed to make sure pre-mRNA species were amplified for RNA editing analysis.*" Therefore, splicing effects are not relevant in our study because we measured the pre-mRNA.

Although not relevant to our experiments, there are plenty of RNA-seq data containing both splicing and editing information available for checking if there's any indication of association between the two processes on a specific editing site. To evaluate whether or not RNA splicing affects editing measurements when total RNA is measured, we checked the GTEx data. We found no significant correlation between splicing ratio and editing level of NEIL1 in any tissue (showing one tissue Adipose_Subcutaneous here as an example). Our analysis suggests that at least for NEIL1, there is no significant confounding effect of RNA splicing on RNA editing.

Fig. RS2-1: Lack of correlation between splicing and editing level for NEIL1. The intron excision ratio was measured for the particular intron in which the ECS of editing site reside, and was processed, filtered and normalized by the GTEx consortium.

As for the RNA half-life, both Reviewer 1 and Reviewer 3 raised the question and acknowledged our response as satisfactory. We looked at the RNA abundance and saw no correlation with editing in general (Supplement Figures 2b, 2e, 2g)

Regarding the response to Reviewer 2, I am not a neuronal network expert, I cannot thus evaluate if what the authors suggest is correct. But I can say that the authors failed to mention in what sense their manuscript has an added value respect the research papers cited: PMID: 26780180, PMID: 29998038, PMID: 28263296. I don't think point 14 is properly addressed either and it is an important point too.

Unless my colleagues say otherwise, I recommend that these points are further addressed before acceptance.

We thank the reviewer for the feedback. There are a few separate points to consider here. The neural network proof-of-concept we presented as a response to reviewer 2's feedback was meant to address the reviewer's concerns about the feasibility of determining sequence and structure features in the absence of a known wild type isoform. Specifically, we were addressing this concern from Reviewer 2:

"Finally, there seems to be little to recommend this modelling strategy as a foundation for developing predictive models of RNA editing – the design of the PREUSS features depend heavily on the WT site, what is the WT site for new sites for transcriptome engineering? What is the WT site when being used detect new editing sites generated by SNPs or somatic mutations?"

Reviewer 2 expressed concern that the featurization we use may be difficult to perform when the wild type isoform sequence is not known, as many of the features are defined in reference to the wild type. We present the neural network proof of concept as a means to predict editing level directly from sequence and structure of an isoform, without an explicit featurization step. We do not include the neural network results in the manuscript itself,

but only in the reviewer response, since we would like to train the neural networks on larger datasets (currently not available) first.

This consideration is separate from the second concern of reviewer 2, where the citations PMID: 26780180, PMID: 29998038, PMID: 28263296 are mentioned. Reviewer 2 states:

“Also, there is an additional question of novelty of the modelling strategy compared to recent work on predicting efficacy of CRISPR guides (PMID: 26780180, PMID: 29998038, PMID: 28263296), this work is not cited in the manuscript and seems to be the most similar work, rather than the splicing code work that is cited.”

These three manuscripts address the design of guide RNAs to minimize off-targets of CRISPR/Cas9 genome editing. As such, they are not relevant to learning sequence and structure determinants of A-to-I RNA editing. We took the suggestion of reviewer 2 previously to cite those papers to suggest that our approach is analogous to the guide design, but not to draw a link to the modeling strategy. To avoid such confusion, we deleted those citations and delete the sentence of *“Our efforts are analogous to the efforts to predict efficiency of CRISPR guides.”*

Regarding the comments about point 14, this is in reference to the following statement from reviewer 2, and our previous response:

“14. Please explain the comment: “Any variant that had more than one mutation was included in the feature matrix twice” – does this mean that each variant is a different training example? If so, this approach seems to generate a bunch of issues with computing performance, stratified sampling for training / test split that are not discussed. Please clarify.”

Yes, each variant constitutes a different training example, as we hypothesized that different combinations of variants within an RNA substrate sequence would produce different effects on editing level. Consequently, a number of the features used for training are based on the position and type of variant. These are documented in Supplementary Table 1 (the “mut_*” categories of features). We verified that there were no cases when an isoform could appear in multiple folds (i.e., no train/test contamination) due to having multiple mutations (all mutations for a given RNA would be exclusively in the training set or exclusively in the validation set or exclusively in the test set). This clarification has been added to the Methods section for clarity. The updated version reads:

“The dataset was randomly separated into 3 splits: training on 70% of variants, model validation on 15%, and testing on the remaining 15%. To avoid train/test contamination, base pair positions along the RNA molecules were assigned to one of the 3 splits (training, tuning, or test). All features associated with a given base pair position were assigned to the corresponding split. Any feature that was null or non-varying across all variants in a given training split was removed from analysis. A number of variants were characterized by two or more mutations relative to the wild type. For these, features were defined for each mutated base pair separately. To avoid train/test contamination, any base pairs that were both mutated in a given variant were included in the same split. The rationale for defining features relative to bases rather than variants is that different combinations of mutations may lead to variations in editing level, and a number of features were derived in reference to mutation type and position (Supplementary Table 1).”

REVIEWERS' COMMENTS

Reviewer #4 (Remarks to the Author):

The authors have addressed my concerns